# A Single Swallow Does Not Make a Summer: Understanding Semantic Structures in Embedding Spaces

## Abstract

Embedding spaces encapsulate rich information from deep learning models, with vector distances reflecting the semantic similarity between textual elements. However, their abstract nature and the computational complexity of analyzing them remain significant challenges. To address these, we introduce the concept of Semantic Field Subspace, a novel mapping that links embedding spaces with the underlying semantics. We propose SAFARI, a novel algorithm for SemAntic Field subspAce deteRmInation, which leverages hierarchical clustering to discover hierarchical semantic structures, using Semantic Shifts to capture semantic changes as clusters merge, allowing for the identification of meaningful subspaces. To improve scalability, we extend Weyl's Theorem, enabling an efficient approximation of Semantic Shifts that significantly reduces computational costs. Extensive evaluations on five real-world datasets demonstrate the effectiveness of SAFARI in uncovering interpretable and hierarchical semantic structures. Additionally, our approximation method achieves a 15~30× speedup while maintaining minimal errors (less than 0.01), making it practical for large-scale applications. The source code is available at https://anonymous.4open.science/r/Safari-C803/.

## 1 Introduction

Embedding spaces are widely recognized for encapsulating rich information learned by deep learning models. In language models, for instance, the distance between embedding vectors often reflects the semantic similarities of corresponding textual elements (Devlin et al., 2019; Wu et al., 2020). However, despite their widespread use, the understanding of embedding spaces remains limited (Ethayarajh, 2019; Clark et al., 2019; Simhi & Markovitch, 2023). Two primary challenges make understanding embedding spaces difficult:

(1) **Abstract Nature:** Embedding spaces reflect complex, high-dimensional relationships between data points, making them inherently abstract. To interpret these spaces, we need a clear connection between the embedded data and their underlying semantics. However, a universally accepted definition of semantics within these spaces remains elusive.

(2) **Computational Complexity:** Understanding embedding spaces requires substantial data, and as these spaces grow richer and more complex, the need for samples increases, straining computational efficiency. Managing this expanding data demands advanced computational resources and optimized algorithms to ensure practical and timely analysis.

Extensive research on embedding spaces spans various perspectives, with two lines of research most relevant to our work: geometry-based and interpretability-based methods. The first line of research focuses on the geometric properties of the embedding space, describing vector distributions and their desired characteristics (Mu & Viswanath, 2018; Liu et al., 2019; Demeter et al., 2020; Ethayarajh, 2019). However, these approaches primarily enhance representation quality by manipulating the geometry of the space, often overlooking the interpretability and semantic coherence of the embeddings. The second line of research concentrates on making the embedding space more interpretable using techniques like rotation, probing (Park et al., 2017; Dufter & Schütze, 2019; Clark et al., 2019; Dalvi et al., 2019), or transforming data into more interpretable dimensions (Simhi & Markovitch, 2023). Despite providing insights into individual dimensions, they often rely on significant assumptions

about the interpretability of original dimensions or require constructing new spaces, which can be computationally intensive and may not preserve the original semantic relationships.

In this paper, we investigate the semantic structure of embedding spaces through textual elements. To address their abstract nature and the challenges of interpretation, we introduce key concepts that link embedding spaces to the underlying semantics. We define each direction as a unique semantic set, serving as a foundation for understanding the structure. Recognizing the context-dependent nature of semantics, we introduce the concept of a Semantic Field for more nuanced interpretation. Identifying Semantic Fields within embedding spaces is framed as an optimization problem. We approximate each Semantic Field as a subspace, referred to as the Semantic Field Subspace, and solve the optimization using Singular Value Decomposition (SVD) (Halko et al., 2009; Trefethen & Bau, 2022).

Building on this concept, we propose SAFARI, a novel algorithm for determining Semantic Field Subspaces through hierarchical clustering. By introducing the concept of Semantic Shift, SAFARI accurately identifies the start and end points of Semantic Field Subspaces within a clustering dendrogram, unveiling the hierarchical structure of Semantic Fields. To overcome the computational challenges of analyzing large datasets, we extend Weyl's Theorem (Weyl, 1912) to approximate Semantic Shift without relying on full SVD, significantly improving computational efficiency.

We evaluate SAFARI on five real-world datasets to validate its ability to uncover hierarchical semantic structures. The results confirm that SAFARI efficiently identifies Semantic Field Subspaces, revealing natural and interpretable hierarchies. Moreover, our approximate method for Semantic Shift computation delivers a $15 \sim 30 \times$ speedup with errors less than 0.01, making it highly practical for large-scale applications. Our contributions are summarized as follows:

(1) We introduce the concept of Semantic Field Subspaces, a novel mapping that bridges embedding spaces with their underlying semantics. It enhances the interpretability of high-dimensional embeddings, facilitating deeper insights into the semantics encoded within vector spaces.

(2) We present SAFARI, a creative algorithm that leverages Semantic Shift to determine Semantic Field Subspaces. By employing hierarchical clustering, SAFARI effectively uncovers the hierarchical semantic structures. We also develop an efficient approximate method for Semantic Shift computation, significantly improving computational efficiency.

(3) We systematically evaluate the efficacy of SAFARI through extensive experiments, demonstrating that our algorithm successfully and efficiently identifies Semantic Filed Subspaces, revealing their hierarchical structures.

## 2 RELATED WORK

**Geometry-based Approaches.** This research focuses on the geometric properties of embedding spaces, aiming to describe the vector distributions and their desired characteristics (Mu & Viswanath, 2018; Liu et al., 2019; Demeter et al., 2020; Ethayarajh, 2019). For instance, Mu & Viswanath (2018) improved word representations by removing the top principal components, while Liu et al. (2019) suppressed transformed dimensions with large variances. Demeter et al. (2020) highlighted how the softmax function weakens geometric structures, introducing bias. A key finding by Ethayarajh (2019) showed that most vectors reside within a narrow cone in the embedding space. Unlike geometry-based research, we do not aim to prove or find the ideal distribution or other geometric properties in the embedding space to improve the model. Instead, SAFARI focuses on revealing and understanding structures within a given embedding space, regardless of its geometric properties. This allows us to maintain the original semantic relationships while uncovering meaningful patterns.

**Interpretability-based Approaches.** This research targets making embedding spaces more interpretable, often through rotation and probing methods (Park et al., 2017; Dufter & Schütze, 2019; Clark et al., 2019; Dalvi et al., 2019). Park et al. (2017) employed rotation algorithms to improve word vector interpretability, while Dufter & Schütze (2019) applied rotation to enhance word space comprehension. Clark et al. (2019) analyzed attention mechanisms in pre-trained models, particularly BERT (Devlin et al., 2019), to gain insights into how the model processes information. Dalvi et al. (2019) examined individual vector dimensions in NLP models to uncover the roles of these dimensions. Recently, Simhi & Markovitch (2023) transformed latent spaces into a new one with more conceptualized and interpretable dimensions. Although SAFARI also seeks to interpret embedding spaces, it differs by not assuming that the original dimensions are inherently interpretable or by

transforming them into new spaces. Instead, SAFARI identify comprehensible structures by linking embedding vectors to their underlying semantic space, preserving the original embeddings and providing a robust framework for semantic interpretation.

# 3 PROBLEM FORMULATION

Before introducing SAFARI, we first define the key concepts of Semantic Distance, Semantic Field, and Semantic Field Subspace. Frequently used notations are summarized in Table A1.

**Semantic Distance.** Let $\mathcal{X}$ be a set of textual elements and $\mathbb{R}^d$ a continuous, $d$-dimensional embedding space learned by a deep learning model $h : \mathcal{X} \to \mathcal{E}$, where $\mathcal{E} \subset \mathbb{R}^d$ denotes embedding vectors for the elements in $\mathcal{X}$. Semantic Distance measures how different two textual elements are, based on the distance between their embedding vectors:

**Definition 1** (Semantic Distance)**:** *The Semantic Distance $d_{sem}(\cdot, \cdot)$ between any two textual elements $\boldsymbol{x}, \boldsymbol{x}' \in \mathcal{X}$ is defined as the cosine distance between their embedding vectors $\boldsymbol{v}, \boldsymbol{v}' \in \mathcal{E}$:*

$$d_{sem}(\boldsymbol{v}, \boldsymbol{v}') = 1 - \langle \boldsymbol{v}, \boldsymbol{v}' \rangle / (\|\boldsymbol{v}\| \cdot \|\boldsymbol{v}'\|). \tag{1}$$

We use $\boldsymbol{x}$ and embedding vector $\boldsymbol{v}$ interchangeably when there is no ambiguity. Each textual element often carries multiple layers of meaning, which we refer to as its semantics. Let $\mathcal{M}$ be the set of all possible semantics. The semantics of an embedding vector $\boldsymbol{v}$ can be expressed as a set $f_{sem}(\boldsymbol{v})$, where $f_{sem}(\boldsymbol{v}) : \mathcal{E} \to 2^{|\mathcal{M}|} \setminus \varnothing$, representing the various semantic facets of $\boldsymbol{v}$.

We argue that *a single textual element $\boldsymbol{x}$ cannot be fully interpreted in isolation*, as its semantics, $f_{sem}(\boldsymbol{v})$, require *context* for interpretation. As the adage states, "You shall know a word by the company it keeps," meaning $\boldsymbol{x}$ becomes interpretable only when considered with related elements in its context. For example, as shown in Fig. 1, the word 'Apple' is ambiguous on its own but gains specific meaning when used in different contexts–referring to a technology company with words like 'Mac,' 'IBM,' and 'Windows,' or to a fruit with words like 'Apple Tree,' 'Juice,' and 'Banana.' The meaning becomes clearer as more contextual words are added.

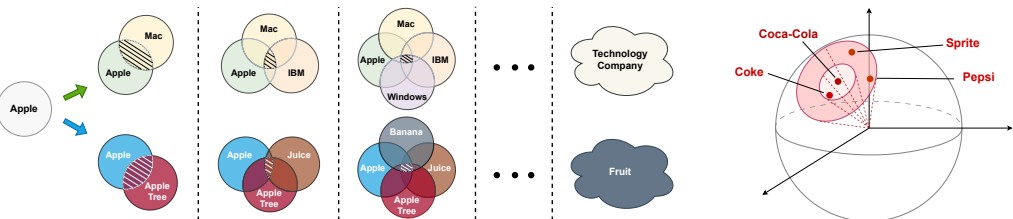

Figure 1: A word 'Apple': from ambiguous to specific.    Figure 2: Close neighborhood.

**Interpreting an Embedding Vector using Close Neighborhood.** As discussed earlier, the semantics of a textual element $\boldsymbol{x}$ can be interpreted through the context in which it appears. In models like Word2Vec (Mikolov et al., 2013) and contextual models like BERT (Devlin et al., 2019), the semantics of an embedding vector $\boldsymbol{v}$ is shaped by the surrounding vectors in its context.

To capture this context, we consider a subset of $\mathcal{E}$ that contains embedding vectors sharing common semantics with $\boldsymbol{v}$. According to Definition 1, these vectors are located near $\boldsymbol{v}$ in terms of Semantic Distance. However, not all nearby vectors contribute meaningful information for interpreting $\boldsymbol{v}$. For instance, as depicted in Fig. 2, the vectors of 'Coca-Cola' and 'Coke' are nearly identical but redundant, as they represent the same concept. Such synonyms are excluded from the interpretation. We approximate synonyms as the $k$-Nearest Neighbors ($k$-NNs) of $\boldsymbol{v}$, typically with $k$ set to 3. The close neighborhood of $\boldsymbol{v}$, denoted as $\mathcal{N}(\boldsymbol{v})$, is defined as:

$$\mathcal{N}(\boldsymbol{v}) = \{\boldsymbol{v}' \mid f_{sem}(\boldsymbol{v}') \cap f_{sem}(\boldsymbol{v}) \neq \varnothing, \boldsymbol{v}' \in \mathcal{E}\} \setminus k\text{-NNs}(\boldsymbol{v}).$$

In Fig. 2, the words 'Sprite' and 'Pepsi' are the close neighborhood of 'Coca-Cola.' Since enumerating the entire set $\mathcal{N}(\boldsymbol{v})$ is impractical, we focus on a subset of $\mathcal{N}(\boldsymbol{v})$ in its context for interpretation.

**Definition 2** (Interpretable Semantics of an Embedding Vector)**:** *Given a subset $\mathcal{N}_{sub}(\boldsymbol{v}) \subseteq \mathcal{N}(\boldsymbol{v})$, the interpretable semantics of $\boldsymbol{v}$ is defined as the intersection of the semantics of all $\boldsymbol{v}' \in \mathcal{N}_{sub}(\boldsymbol{v})$:*

$$f_{int}(\boldsymbol{v}) = \bigcap_{\boldsymbol{v}' \in \mathcal{N}_{sub}(\boldsymbol{v})} f_{sem}(\boldsymbol{v}'). \tag{2}$$

**Semantic Field.** After interpreting the semantics of a single vector $\boldsymbol{v}$, we extend this to a set of embedding vectors $\mathcal{C} \subseteq \mathcal{E}$, referred to as a Semantic Field. Similar to Definition 2, the Semantic Field captures the shared semantics across multiple vectors by intersecting the meanings of vectors in $\mathcal{C}$. Since $\mathcal{C}$ may contain vectors with varying semantics, we refine it to include only those that share the most common semantics. Formally, we define the Semantic Field as:

**Definition 3** (Semantic Field)**:** *Given a set $\mathcal{C}$ of embedding vectors, the Semantic Field is defined as:*

$$F_{int}(\mathcal{C}) = \bigcap_{\boldsymbol{v} \in \mathcal{C}^*} f_{sem}(\boldsymbol{v}), \tag{3}$$

*where $\mathcal{C}^*$ is a subset of $\mathcal{C}$ that maximizes shared semantics by minimizing the symmetric difference:*

$$\mathcal{C}^* = \arg\min_{\mathcal{C}_{sub} \subseteq \mathcal{C}} \left| \bigcup_{\boldsymbol{v}' \in \mathcal{C}_{sub}} f_{sem}(\boldsymbol{v}') - \bigcap_{\boldsymbol{v}' \in \mathcal{C}_{sub}} f_{sem}(\boldsymbol{v}') \right|. \tag{4}$$

Suppose $\mathcal{C}_{sub} = \{\boldsymbol{v}_1, \boldsymbol{v}_2\}$. As shown in Fig. 3, the symmetric difference is visualized as the shadow area, and minimizing it helps identify the optimal subset $\mathcal{C}^* \subseteq \mathcal{C}$ that shares the most common semantics (the overlapping area). Nevertheless, the concept of a Semantic Field relies on the latent semantic function $f_{sem}(\cdot)$, making them abstract and hard to compute directly. To remedy this issue, we introduce the concept of a Semantic Field Subspace.

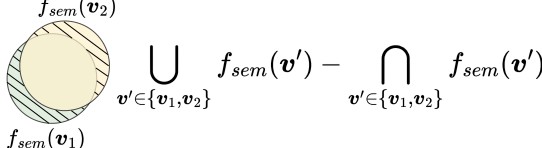

Figure 3: Visualization of symmetric difference.

**Semantic Field Subspace.** According to Definition 1, embedding vectors pointing in the same directions represent the same semantics. This insight leads to the idea that *every vector in the embedding space corresponds to a unique set of semantics*, and conversely, a specific vector can represent each semantic set. Hence, a Semantic Field can be approximated by treating the semantic sets as embedding vectors and solving the optimization in Eq. (4).

**Definition 4** (Semantic Field Subspace)**:** *Let $\mathbb{S}$ be a subspace of $\mathbb{R}^d$. The semantics of $\mathbb{S}$ is defined as:*

$$F_{sem}(\mathbb{S}) = \bigcap_{\boldsymbol{v} \in \mathcal{C}^*} f_{sem}(\boldsymbol{v}), \tag{5}$$

*where $\mathcal{C}^*$ is the set of embedding vectors in $\mathbb{S}$ that minimizes the following symmetric difference:*

$$\mathcal{C}^* = \arg\min_{\mathcal{C} \subset \mathbb{S}} \left| \bigcup_{\boldsymbol{v}' \in \mathcal{C}} f_{sem}(\boldsymbol{v}') - \bigcap_{\boldsymbol{v}' \in \mathcal{C}} f_{sem}(\boldsymbol{v}') \right|. \tag{6}$$

Although enumerating all vectors in a subspace is still impractical, we can approximate a subspace using a finite set of representative vectors $\mathcal{C} \subseteq \mathcal{E}$. Representing $\mathcal{C}$ as a matrix $\boldsymbol{A}$ allows us to apply SVD to extract the key components, capturing the essential semantics of the subspace. We define this relaxed subspace approximation as follows:

**Definition 5** (Relaxed Version of Semantic Field Subspace)**:** *Given a subset $\mathcal{C} \subseteq \mathcal{E}$ represented by a matrix $\boldsymbol{A}$, the subspace $\mathbb{S}$ can be approximated by $\boldsymbol{A}$, i.e., $\mathbb{S} \approx \boldsymbol{A}$. Applying SVD to $\boldsymbol{A}$ gives $\boldsymbol{A} = \boldsymbol{U}\boldsymbol{\Sigma}\boldsymbol{V}^\top$, and the Semantic Field Subspace $F_{sem}(\mathbb{S})$ is approximated by the singular values in $\boldsymbol{\Sigma}$ and the singular vectors in $\boldsymbol{V}^\top$, i.e., $F_{sem}(\mathbb{S}) \approx \boldsymbol{\Sigma}, \boldsymbol{V}^\top$.*

## 4 METHODOLOGY

### 4.1 THE SAFARI ALGORITHM

**Motivation.** SAFARI is designed to identify Semantic Field Subspaces from a set $\mathcal{E}$ of embedding vectors. While Definition 5 provides a theoretical framework for constructing these subspaces, the primary challenge lies in selecting appropriate subsets $\mathcal{C} \subseteq \mathcal{E}$, especially in a large, diverse text corpus. SAFARI addresses this by leveraging the natural clustering property of embedding vectors, where clusters represent specific topics or semantic themes, enabling efficient identification of meaningful subspaces in complex, high-dimensional embedding spaces.

**Algorithm Description.** SAFARI utilizes hierarchical clustering to iteratively identify potential Semantic Field Subspaces, as it offers flexibility by not requiring a predefined number of clusters, allowing for dynamic exploration of the data. Additionally, hierarchical clustering produces a dendrogram, which helps interpret relationships between clusters and their semantic structures. The pseudo-code for SAFARI is shown in Algorithm 1.

---

**Algorithm 1:** SAFARI

---

**Input:** A set $\mathcal{E}$ of embedding vectors in $\mathbb{R}^d$, window size $w$;
**Output:** A set $\Psi$ of clusters with specific Semantic Field Subspaces;
1   $\Phi \leftarrow$ Initialize each $\boldsymbol{v} \in \mathcal{E}$ as its own cluster;
2   $iter = 0; \mu = 0; \tau = 0;$
3   $\Psi \leftarrow \varnothing;$     ▷ Store clusters with specific Semantic Field Subspaces
4   **while** $|\Phi| > 1$ **do**
     ▷ Step 1: Cluster Merging
5      $\{\mathcal{C}_x, \mathcal{C}_y\} \leftarrow \arg\min_{\mathcal{C}_i, \mathcal{C}_j \in \Phi} d_{sem}(\mathcal{C}_i, \mathcal{C}_j);$
6      $\mathcal{C}_{new} \leftarrow \mathcal{C}_x \cup \mathcal{C}_y;$
7      $\Phi \leftarrow \Phi \cup \mathcal{C}_{new} \setminus \{\mathcal{C}_x, \mathcal{C}_y\};$
     ▷ Step 2: Semantic Field Subspace Determination
8      $\mathcal{C}_x \leftarrow |\mathcal{C}_x| > |\mathcal{C}_y| ? \mathcal{C}_x : \mathcal{C}_y;$
9      Compute the Semantic Shift $\Delta F_{sem}(\mathcal{C}_x, \mathcal{C}_{new})$ using Algorithm 2;
10      **if** $\Delta F_{sem}(\mathcal{C}_x, \mathcal{C}_{new}) > \mu + 2\tau$ **then** $\Psi \leftarrow \Psi \cup \mathcal{C}_{new};$
11      $iter = iter + 1;$
12      Update $\mu$ and $\tau$ by considering $\Delta F_{sem}(\mathcal{C}_x, \mathcal{C}_{new})$ and the previous $(w - 1)$ values;
13   **return** $\Psi$;

---

Initially, each embedding vector in $\mathcal{E}$ forms its own cluster, resulting in $n$ clusters (Line 1). These clusters do not yet provide meaningful interpretation or form Semantic Field Subspaces. An empty set $\Psi$ is initialized to store clusters with specific semantic meanings (Line 3). The algorithm then iterates through two key steps until all clusters are merged into a single one (Lines 4–12).

- **Step 1: Cluster Merging.** First, the two closest clusters, $\mathcal{C}_x$ and $\mathcal{C}_y$, are identified based on the Semantic Distance $d_{sem}(\mathcal{C}_x, \mathcal{C}_y)$, with the centroid representing each cluster (Line 5). These two clusters are then merged into a new cluster $\mathcal{C}_{new}$ (Line 6), after which the original clusters $\mathcal{C}_x$ and $\mathcal{C}_y$ are removed, and the new cluster $\mathcal{C}_{new}$ is added to the set $\Phi$ (Line 7).
- **Step 2: Semantic Field Subspace Determination.** The larger cluster, $\mathcal{C}_x$, is selected, and the Semantic Shift $\Delta F_{sem}(\mathcal{C}_x, \mathcal{C}_{new})$ is computed to measure the semantic gap between $\mathcal{C}_x$ and $\mathcal{C}_{new}$ (Lines 8–9), where its definition and computation will be presented later. A sliding window of size $w$ tracks the last $w$ Semantic Shift values, calculating their mean ($\mu$) and standard deviation ($\tau$). If $\Delta F_{sem}(\mathcal{C}_x, \mathcal{C}_{new})$ exceeds the *dynamic* threshold ($\mu + 2\tau$), indicating a large semantic gap, $\mathcal{C}_{new}$ is added to $\Psi$ as a Semantic Field Subspace (Line 10). At last, the algorithm updates the iteration counter and recalculates $\mu$ and $\tau$ for the next iteration (Lines 11–12).

**Exact Semantic Shift Computation.** We now define and describe the process for computing the Semantic Shift (pseudo-code provided in Appendix B). Given two clusters, $\mathcal{C}_x$ and $\mathcal{C}_{new}$, we first construct matrices $\boldsymbol{A}_x$ and $\boldsymbol{A}_{new}$, representing their respective subspaces, $\mathbb{S}_x$ and $\mathbb{S}_{new}$. Following Definition 5, SVD is performed on these two matrices $\boldsymbol{A}_x$ and $\boldsymbol{A}_{new}$ to approximate the semantics of these subspaces: $F_{sem}(\mathbb{S}_x) \approx \boldsymbol{\Sigma}_x, \boldsymbol{V}_x^\top$ and $F_{sem}(\mathbb{S}_{new}) \approx \boldsymbol{\Sigma}_{new}, \boldsymbol{V}_{new}^\top$.

We then compare the singular vectors $\boldsymbol{v}_i \in \boldsymbol{V}_x^\top$ with their nearest neighbors $\tilde{\boldsymbol{v}}_i^* \in \boldsymbol{V}_{new}^\top$, based on Semantic Distance $d_{sem}(\boldsymbol{v}_i, \tilde{\boldsymbol{v}}_i^*)$, which captures shifts in semantic direction. For each singular value $\sigma_i \in \boldsymbol{\Sigma}_x$ and $\tilde{\sigma}_i \in \boldsymbol{\Sigma}_{new}$ sorted in descending order, we calculate the difference $\Delta\sigma_i = |\sigma_i - \tilde{\sigma}_i|$, reflecting shifts in the importance of each dimension. Thus, the total Semantic Shift between clusters $\mathcal{C}_x$ and $\mathcal{C}_{new}$ (or subspaces $\mathbb{S}_x$ and $\mathbb{S}_{new}$) is defined as:

$$\Delta F_{sem}(\mathcal{C}_x, \mathcal{C}_{new}) = \Delta F_{sem}(\mathbb{S}_x, \mathbb{S}_{new}) = \sum_i \Delta\sigma_i \cdot d_{sem}(\boldsymbol{v}_i, \tilde{\boldsymbol{v}}_i^*). \tag{7}$$

Eq. (7) captures both the *importance difference* (through $\Delta\sigma_i$) and *directional difference* (through $d_{sem}(\boldsymbol{v}_i, \tilde{\boldsymbol{v}}_i^*)$), providing a comprehensive measure of the semantic gap between subspaces.

**Example 1:** Consider a toy example with 11 words. The dendrogram in Fig. 4 shows how Algorithm 1 identifies Semantic Field Subspaces through hierarchical clustering. In the first three iterations, semantically similar word pairs, such as 'Macbook Air' and 'Macbook Pro,' 'PowerPoint' and 'Excel,' and 'Michael Jordan' and 'Chicago Bulls,' are merged together. These merges result in only minor Semantic Shifts, so they are not recognized as Semantic Field Subspaces. However, in the 4th iteration, the word 'Apple' is merged with the 'Macbook Air' and 'Macbook Pro' cluster, which produces a significant Semantic Shift, indicating the creation of a new Semantic Field Subspace.

This example also highlights the hierarchical nature of Semantic Field Subspaces. For instance, the 'IT Companies' subspace encompasses both 'Apple (as an IT Company)' and 'Microsoft (as an IT Company)' as individual subspaces within it. Additionally, as the clustering process continues, SAFARI dynamically tracks Semantic Shifts using a sliding window, allowing it to adjust the threshold in real-time. This adaptive mechanism ensures that irrelevant clusters (like 'IT Companies' and 'NBA') do not form new Semantic Field Subspaces, preserving semantic integrity. △

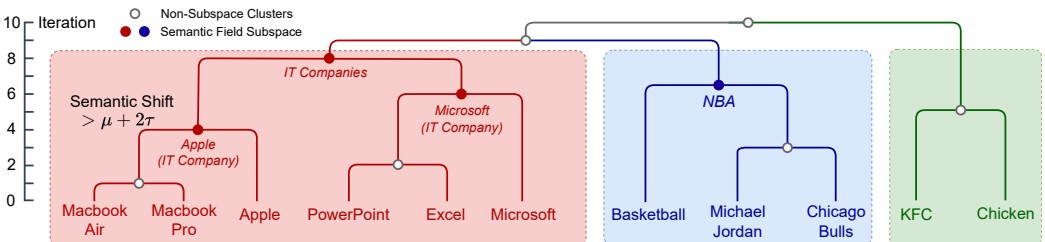

Figure 4: A toy example of Algorithm 1.

### 4.2 APPROXIMATE SEMANTIC SHIFT COMPUTATION AND THEORETICAL ANALYSIS

**Approximate Semantic Shift Computation.** Exact Semantic Shift computation is computationally intensive due to the need for a full SVD on two dense matrices, $\boldsymbol{A}_x$ and $\boldsymbol{A}_{new}$. For a matrix of size $n \times d$ ($d \le n$), the SVD has a time complexity of $O(nd^2)$ (Halko et al., 2009; Trefethen & Bau, 2022), which becomes prohibitive when repeated at each iteration. To address this, we develop an efficient approximation algorithm (pseudo-code in Appendix C). Given the larger cluster $\mathcal{C}_x$ and the smaller cluster $\mathcal{C}_y$, we first construct matrices $\boldsymbol{A}_x$ and $\boldsymbol{A}_y$. We then compute the spectral norm of $\boldsymbol{A}_y$ and the maximum singular value $\sigma_{max}$ from $\boldsymbol{A}_x$, using them to approximate the Semantic Shift of $\mathcal{C}_x$ and $\mathcal{C}_{new}$:

$$\Delta \tilde{F}_{sem}(\mathcal{C}_x, \mathcal{C}_{new}) = \|\boldsymbol{A}_y\|_2 \, \sigma_{max}(\boldsymbol{A}_x). \tag{8}$$

**Theoretical Analysis.** Next, we provide a theoretical analysis to show that the Semantic Shift (Eq. (7)) can be accurately approximated by Eq. (8). Given two matrices $\boldsymbol{A}_x$ and $\boldsymbol{A}_y$, let $\boldsymbol{A}_{new} = [\boldsymbol{A}_x | \boldsymbol{A}_y]$ be the matrix of the newly merged cluster $\mathcal{C}_{new}$. We begin with Theorem 1, which shows that the magnitude difference ($\Delta \sigma_i$) between $\boldsymbol{A}_x$ and $\boldsymbol{A}_{new}$ is bounded.

**Theorem 1:** *Given any two matrices $\boldsymbol{A}_x$ and $\boldsymbol{A}_y$ with equal number of columns, where $\boldsymbol{A}_x$ has more rows than $\boldsymbol{A}_y$, the following inequality holds:*

$$\Delta \sigma_i = |\sigma_i(\boldsymbol{A}_x) - \sigma_i(\boldsymbol{A}_{new})| \le \|\boldsymbol{A}_y\|_2. \tag{9}$$

*Proof.* The proof relies on Weyl's Theorem (Weyl, 1912), which we first present. In each iteration where two clusters are merged, let $\boldsymbol{A} \in \mathbb{R}^{m \times d}$ denote the matrix of the larger cluster. The merging process introduces a perturbation to $\boldsymbol{A}$, denoted as $\boldsymbol{E}$, and the perturbed matrix is given by $\tilde{\boldsymbol{A}} = \boldsymbol{A} + \boldsymbol{E}$. Weyl's Theorem provides a bound on the change in singular values caused by this perturbation:

**Theorem 2** (Weyl's Theorem (Weyl, 1912))**:**

$$|\sigma_i(\boldsymbol{A}) - \sigma_i(\tilde{\boldsymbol{A}})| = |\sigma_i(\boldsymbol{A}) - \sigma_i(\boldsymbol{A} + \boldsymbol{E})| \le \|\boldsymbol{E}\|_2. \tag{10}$$

Weyl's Theorem states that the singular values of a matrix cannot change by more than the spectral norm of the perturbation matrix $\boldsymbol{E}$. Even though $\boldsymbol{E}$ is often assumed to be small in matrix perturbation theory, this result holds for any perturbation, regardless of the size of $\|\boldsymbol{E}\|_2$ (Stewart, 1998).

We now apply Weyl's Theorem to prove Theorem 1. Let $\boldsymbol{O}$ be a zero matrix. Thus, $\boldsymbol{A}_{new} = [\boldsymbol{A}_x | \boldsymbol{A}_y] = [\boldsymbol{A}_x | \boldsymbol{O}] + [\boldsymbol{O} | \boldsymbol{A}_y]$. According to Theorem 2, we have:

$$|\sigma_i([\boldsymbol{A}_x | \boldsymbol{O}]) - \sigma_i(\boldsymbol{A}_{new})| \le \|[\boldsymbol{O} | \boldsymbol{A}_y]\|_2 = \|\boldsymbol{A}_y\|_2.$$

To complete the proof, we need to show that $\sigma_i([\boldsymbol{A}_x | \boldsymbol{O}]) = \sigma_i(\boldsymbol{A}_x)$. Since this equality always holds, Theorem 1 is proved. □

Based on Theorem 1, Eq. (7) can be rewritten as: $\Delta F_{sem}(\mathcal{C}_x, \mathcal{C}_{new}) = \sum_i \Delta \sigma_i \cdot d_{sem}(\boldsymbol{v}_i, \tilde{\boldsymbol{v}}_i^*) \le \sum_i \|\boldsymbol{A}_y\|_2 \cdot d_{sem}(\boldsymbol{v}_i, \tilde{\boldsymbol{v}}_i^*) = \|\boldsymbol{A}_y\|_2 \cdot \sum_i d_{sem}(\boldsymbol{v}_i, \tilde{\boldsymbol{v}}_i^*)$. To further approximate $\sum_i d_{sem}(\boldsymbol{v}_i, \tilde{\boldsymbol{v}}_i^*)$, we present the following theorem:

**Theorem 3:** *Given two matrices $A_x$ and $A_y$ used in the exact Semantic Shift computation as presented in Section 4.1, the directional difference between them is proportional to the largest singular vector $\sigma_{max}$ of $A_x$:*

$$\sum_i d_{sem}(v_i, \tilde{v}_i^*) = \mathcal{O}(\sigma_{max}(A_x)). \tag{11}$$

*Proof.* To prove Theorem 3, we first introduce Lemma 1.

**Lemma 1:** *For a given matrix $M$ with a series of singular values $\{\sigma_i\}$, the condition number is $\kappa(M) = \frac{\sigma_{max}}{\sigma_{min}}$. The condition number quantifies the matrix's sensitivity to small perturbations, where higher values indicate greater susceptibility to changes (Belsley et al., 2005; Meyer, 2023). Consequently, we establish:*

$$\sum_i d_{sem}(v_i, \tilde{v}_i^*) = \mathcal{O}(\kappa(M)). \tag{12}$$

Given that real-world matrices contain noise and are often rank-deficient, we assume that:

$$\forall M, \lim_{i \to r} \sigma_i = 0,$$
$$\forall M_1, M_2, \lim_{i \to r_1} \sigma_i(M_1) = \mathcal{O}(\lim_{j \to r_2} \sigma_j(M_2)). \tag{13}$$

Using this assumption, we compare the condition numbers of two matrices $M_1$ and $M_2$:

$$\frac{\kappa(M_1)}{\kappa(M_2)} = \frac{\sigma_{max}(M_1) \times \sigma_{min}(M_2)}{\sigma_{max}(M_2) \times \sigma_{min}(M_1)}. \tag{14}$$

According to Eq. (13), we have $\frac{\kappa(M_1)}{\kappa(M_2)} = \frac{\sigma_{max}(M_1)}{\sigma_{max}(M_2)}$. Thus, comparing condition numbers of matrices under this assumption (Eq. (13)) is equivalent to comparing their largest singular values. Therefore,

$$\kappa(M) = \mathcal{O}(\sigma_{max}(M)). \tag{15}$$

With Lemma 1, we establish that the condition number, and thus the total directional difference, can be approximated by the largest singular value. This completes the proof of Theorem 3. □

By utilizing this approximation for Semantic Shifts, we can bypass the need for full SVD in each iteration. Instead, we only need to compute $\|A_y\|_2$ and $\sigma_{max}(A_x)$, significantly reducing the time complexity in Algorithm 1 (Meyer, 2023; Horn & Johnson, 2012).

## 5 EXPERIMENTS

### 5.1 SEMANTIC FIELD SUBSPACE ISOLATION

**Experiment Setup.** We first evaluate whether the Semantic Field Subspaces determined by SAFARI effectively preserve their semantic meanings and remain isolated from one another. We employ four datasets: AG-News (Zhang et al., 2015), AAPD (Yang et al., 2018), IMDB (Maas et al., 2011), and Yelp[1] (see Appendix D for details). Using BLINK (Wu et al., 2020) for entity linking, we extract entities from each dataset and rank them by their TF-IDF scores (Schütze et al., 2008; Leskovec et al., 2020), selecting the top 10%. These entities are then split into 80% for training (used to identify subspaces) and 20% for testing. For each Semantic Field Subspace, we retain the top 100 singular vectors. We then compute the average distance between test entities and the identified subspaces to evaluate the isolation and preservation of semantic meaning. The results are presented in Fig. 5.

**Result Analysis.** The results in Fig. 5 yield three key observations: (1) Test entities are closest to the subspace corresponding to their respective dataset and show reasonable distances to others, confirming that SAFARI effectively preserves semantic meanings within isolated subspaces. (2) Entities from AAPD exhibit the greatest distance from other subspaces, reflecting the significant semantic gap between academic papers and other types of content. (3) Entities within AAPD are further from their own subspace compared to other datasets, likely due to the smaller number of entities extracted from academic papers, resulting in a less rich semantic representation.

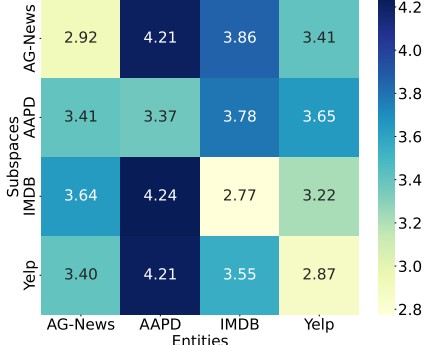

Figure 5: Semantic Field Subspace Isolation.

[1] https://www.yelp.com/dataset

## 5.2 SEMANTIC FIELD SUBSPACE CLASSIFICATION

**Experiment Setup.** We conduct a classification experiment to further assess whether the Semantic Field Subspaces identified by SAFARI retain semantic meaning. AG-News is divided into four categories: Business, Sci/Tech, Sports, and World, while AAPD, IMDB, and Yelp are grouped based on their content: academic papers, movie reviews, and business entities, respectively, resulting in 7 distinct categories (classes). Following the setup from Section 5.1, we assign class labels to entities and select the top-$n$ entities from each class, using 80% for training and 20% for testing. Semantic Field Subspaces are constructed using training data, with each subspace assigned a class label. For the test entities, we calculate the distance (weighted by singular values) to all subspaces, predicting the label of the nearest one. We compare the performance of SAFARI against several baselines, including SVM (Platt, 1999; Chang & Lin, 2011), KNN (Cover & Hart, 1967; Fix, 1985), Random Forest (Breiman, 2001), and BERT (Dalvi et al., 2019), with Random Guess as a trivial baseline. Classification accuracy and training time are presented in Figs. 6 and 7.

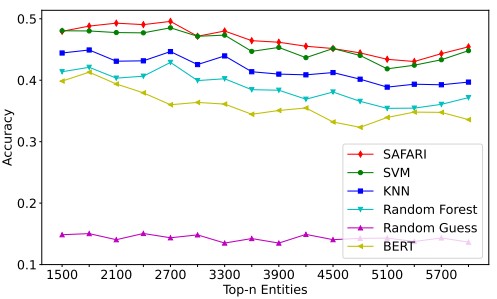

Figure 6: Classification accuracy.

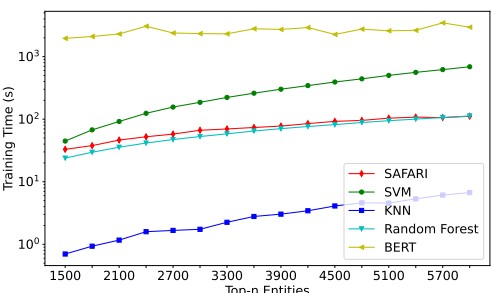

Figure 7: Training time.

**Result Analysis.** The results in Fig. 6 reveals three crucial findings: (1) SAFARI and SVM achieve the highest accuracy, with SAFARI slightly outperforming in some cases. (2) KNN and Random Forest perform moderately well, both surpassing BERT and Random Guess. (3) BERT lags in accuracy due to its reliance on embedding vectors rather than raw text. Regarding efficiency (see Fig. 7), BERT incurs the highest time cost, increasing sharply as the number of entities grows. SVM also shows a steep rise in time cost but remains much faster than BERT. In contrast, SAFARI and Random Forest maintain consistently low time costs. Overall, SAFARI strikes the best balance between accuracy and efficiency, confirming its efficacy in retaining semantic meaning for classification.

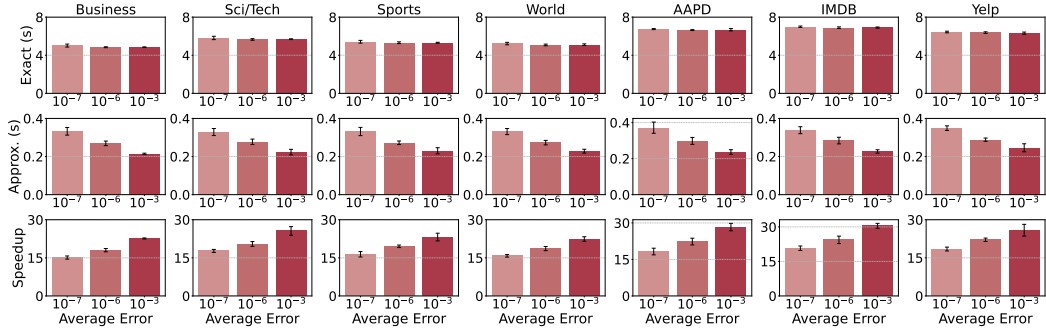

Figure 8: Efficiency comparison across 7 distinct classes.

## 5.3 EFFICIENCY

**Experiment Setup.** We then assess the speed of Semantic Shift computation by comparing full SVD with our approximation method. The input data includes the top 2,000 entities from each class, processed via hierarchical clustering, with the resulting matrices used to measure computational efficiency. The results are averaged over 10 independent runs.

**Result Analysis.** As shown in Fig. 8, our approximation method is significantly faster than full SVD, achieving speedups of 15∼30× across 7 distinct classes. The small variations, indicated by error bars, reflect stability across runs. While the speedup increases with larger errors, we maintained at

$10^{-3}$, suggesting the potential for even greater speedups with larger errors. These results underscore the efficiency and reliability of our approximation method for computing Semantic Shifts.

## 5.4 HIERARCHICAL SEMANTIC STRUCTURE

**Experiment Setup.** We validate SAFARI's ability to uncover hierarchical semantic structures by applying it to the top 1,000 entities from each category, with a focus on the Sports category from the AG-News dataset. The Sports category was chosen due to its well-structured, event-driven content, providing an ideal setting for evaluation. At each iteration, Semantic Shifts are computed to identify Semantic Field Subspaces, using both exact and approximation methods to further confirm SAFARI's ability to deliver accurate approximations in real-world data. The results from iteration 11,000 to 16,000 are depicted in Fig. 9, with specific shifts at iterations 11,352 and 15,856 highlighted to showcase the hierarchical structure, further analyzed in Figs. 10 and 11, respectively.

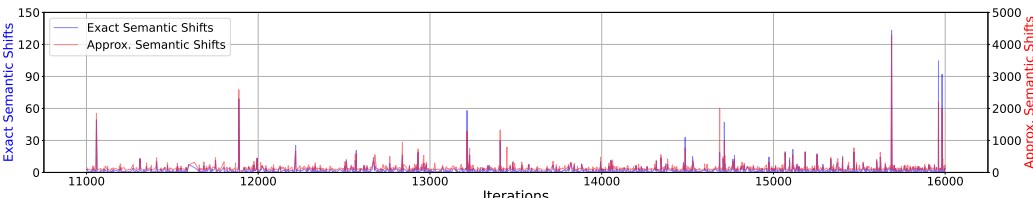

Figure 9: Exact and approximate Semantic Shift values for the Sports category in AG-News.

**Result Analysis.** First, SAFARI shows high accuracy in approximating Semantic Shifts. As displayed in Fig. 9, the approximate curve (red) closely aligns with the exact curve (blue), with a high Pearson correlation coefficient of **0.92**, confirming the effectiveness of our approximation method outlined in Section 4.2. Moreover, the dynamic thresholding mechanism in SAFARI, using a sliding window, effectively captures the Semantic Field Subspaces with smaller Semantic Shifts that might otherwise be missed due to varying Semantic Shift magnitudes.

Second, SAFARI captures hierarchical relationships with varying granularity. In Fig. 10, small initial clusters (e.g., individual USA university basketball or football teams) gradually evolve into broader categories like "University Football and Basketball Teams from the USA." Similarly, Fig. 11 shows how teams initially grouped by country later form broader regional clusters, with European teams (blue labels) grouping more closely than non-European teams (read labels). This progression highlights SAFARI's ability to preserve semantic relationships while uncovering the hierarchical structure within the data. Further analysis is provided in Appendix E.1.

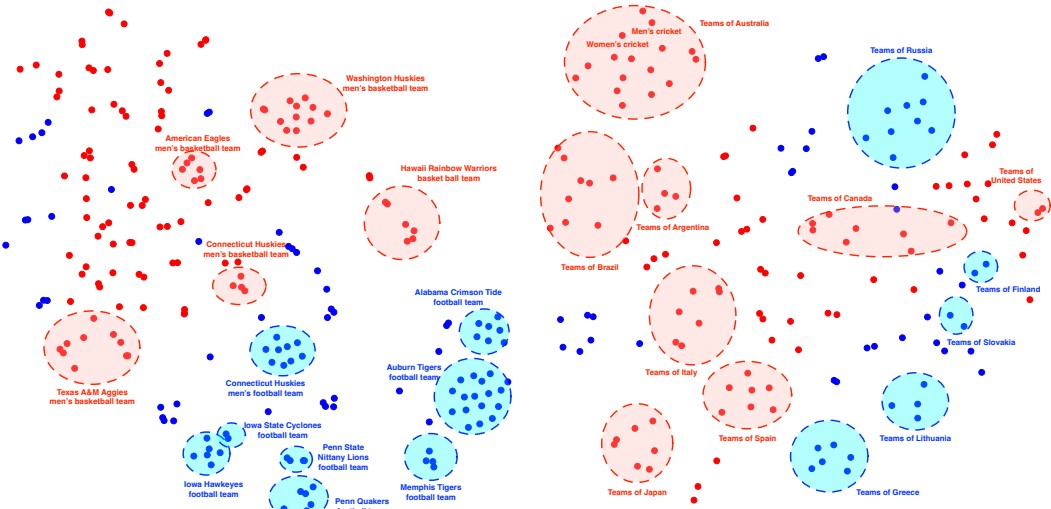

Figure 10: USA basketball teams merged with USA football teams.

Figure 11: Sports teams from different locations merged.

## 5.5 FAKE NEWS EXPLANATION

**Experiment Setup.** Finally, we showcase how SAFARI can provide detailed explanations using the FakeNewsCorpus dataset (Pathak & Srihari, 2019), which divides fake news articles into ten categories: Bias, Clickbait, Conspiracy, Fake, Hate, JunkSci, Political, Reliable, Satire, and Unreliable. Detailed descriptions are provided in Table D2. Unlike content-based labels in previous datasets, these domain-based labels necessitate deeper explanations to clarify classification reasoning. For example, an article labeled JunkSci might overlap with the Clickbait category, emphasizing the need for precise explanations to avoid ambiguity. This complexity makes this dataset more challenging, showcasing SAFARI's ability to handle nuanced and overlapping categories.

We extract entities from each news article and construct ten Semantic Field Subspaces. By comparing these subspaces, we analyze their principal directions and perform a nearest neighbor search to identify the top $k$ Wikipedia entities. To enhance interpretability, each entity is mapped to a Wikidata node (Vrandečić & Krötzsch, 2014), where we examine the associated labels (Ayoola et al., 2022), and the top node types are grouped into broader categories, offering insights into the classification and relationships between the news article categories.

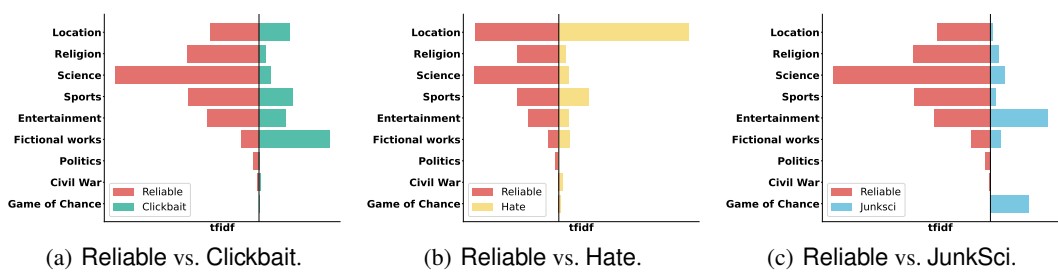

(a) Reliable vs. Clickbait.  (b) Reliable vs. Hate.  (c) Reliable vs. JunkSci.

Figure 12: Reliable news compared with Clickbait, Hate, and JunkSci news.

**Result Analysis.** The Semantic Field Subspaces constructed by SAFARI for the ten fake news categories display distinct patterns (see Appendix E.2 for details). To explain these differences, we compare each subspace with the Reliable subspace, focusing on three specific categories: Clickbait, Hate, and JunkSci (see Fig. 12). Further comparison results are available in Appendix E.3.

- **Reliable**: Reliable news consistently scores high for node types related to *Science*, *Religion*, and *Sports*. This suggests that fake news sources prioritize emotionally charged and controversial topics, while Reliable sources focus more on factual, less sensational content.
- **Clickbait**: Articles labeled as Clickbait show high values for the *Fictional works* node type, alongside strong associations with *Entertainment* and *Sports* (see Fig. 12(a)). This suggests that clickbait content often revolves around sensational or fictional topics to capture attention, with less emphasis on scientific or factual information.
- **Hate**: The Hate category emphasizes the *Location* node type, reflecting its frequent associations with political propaganda and regional tensions (see Fig. 12(b)). This geographic specificity is distinct to Hate news, as it amplifies political or cultural division based on location.
- **JunkSci**: JunkSci sources are linked to the *Game of Chance* and *Entertainment* node types, suggesting a focus on random outcomes and unscientific topics (see Fig. 12(c)). This aligns with the nature of junk science, as it often lacks scientific rigor, unlike reliable sources that emphasize factual content such as science and religion.

## 6 CONCLUSIONS

In this paper, we tackled the challenge of understanding the abstract and intricate structure of embedding spaces. We introduced SAFARI, a novel algorithm for identifying Semantic Field Subspaces through hierarchical clustering and the concept of Semantic Shift, with an efficient approximation method that avoids SVD, reducing the computational cost. Extensive experiments on five real-world datasets demonstrated SAFARI's capability to uncover interpretable, hierarchical semantic structures while achieving substantial computational savings. SAFARI has shown to be effective for tasks like classification and explanation. This work not only bridges the gap between embedding spaces and their underlying semantics but also offers new insights into the structure and utility of embedding spaces, enhancing both interpretability and efficiency in their analysis.

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

## A  TABLE OF NOTATIONS

Table A1: List of frequently used notations.

| Symbol | Description |
|---|---|
| $\boldsymbol{x}, \mathcal{X}$ | A set $\mathcal{X}$ of textual elements $\boldsymbol{x}$, i.e., $\boldsymbol{x} \in \mathcal{X}$ |
| $\boldsymbol{v}, \mathcal{E}$ | A set $\mathcal{E}$ of embedding vectors $\boldsymbol{v}$ in a $d$-dimensional embedding space $\mathbb{R}^d$, i.e., $\boldsymbol{v} \in \mathcal{E}$ and $\mathcal{E} \subset \mathbb{R}^d$ |
| $h$ | A deep learning model: $\mathcal{X} \to \mathcal{E}$ |
| $d_{sem}(\boldsymbol{v}, \boldsymbol{v}')$ | The semantic distance between any two embedding vectors $\boldsymbol{v}$ and $\boldsymbol{v}'$ |
| $\mathcal{M}$ | A semantic set |
| $\mathcal{N}(\boldsymbol{v})$ | The close neighborhood of an embedding vector $\boldsymbol{v}$ |
| $f_{sem}(\boldsymbol{v})$ | The semantics of an embedding vector $\boldsymbol{v}$, i.e., $f_{sem}(\boldsymbol{v}) : \mathbb{R}^d \to 2^{|\mathcal{M}|} \setminus \{\varnothing\}$ |
| $f_{int}(\boldsymbol{v})$ | The interpretable semantics of an embedding vector $\boldsymbol{v}$, i.e., $f_{int}(\boldsymbol{v}) : \mathbb{R}^d \to 2^{|\mathcal{M}|} \setminus \{\varnothing\}$ |
| $F_{int}(\mathcal{C})$ | The semantic field of a subset of embedding vectors $\mathcal{C} \subseteq \mathcal{E}$, i.e., $F_{int}(\mathcal{C}) : \mathbb{R}^d \to 2^{|\mathcal{M}|} \setminus \{\varnothing\}$ |
| $F_{sem}(\mathbb{S})$ | The semantic field subspace of a subspace $\mathbb{S} \subseteq \mathbb{R}^d$, i.e., $F_{sem}(\mathbb{S}) : 2^{|\mathbb{R}^d|} \setminus \{\varnothing\} \to 2^{|\mathcal{M}|} \setminus \{\varnothing\}$ |
| $\Delta F_{sem}(\mathcal{C}_1, \mathcal{C}_2)$ | The semantic shift of any two clusters $\mathcal{C}_1$ and $\mathcal{C}_2$ |

## B  PSEUDO-CODE FOR EXACT SEMANTIC SHIFT COMPUTATION

The pseudo-code for exact Semantic Shift computation is depicted in Algorithm 2.

---
**Algorithm 2:** Exact Semantic Shift Computation

---
**Input:** Larger cluster $\mathcal{C}_x$ and new cluster $\mathcal{C}_{new}$;
**Output:** Exact Semantic Shift $\Delta F_{sem}(\mathcal{C}_x, \mathcal{C}_{new})$;
1  $\boldsymbol{A}_x$ and $\boldsymbol{A}_{new} \leftarrow$ Construct two matrices from $\mathcal{C}_x$ and $\mathcal{C}_{new}$;
2  $F_{sem}(\mathbb{S}_x) \approx \boldsymbol{\Sigma}_x, \boldsymbol{V}_x^\top$ and $F_{sem}(\mathbb{S}_{new}) \approx \boldsymbol{\Sigma}_{new}, \boldsymbol{V}_{new}^\top \leftarrow$ Perform SVD on $\boldsymbol{A}_x$ and $\boldsymbol{A}_{new}$;
3  Compute $\Delta F_{sem}(\mathcal{C}_x, \mathcal{C}_{new})$ using Eq. (7);
4  **return** $\Delta F_{sem}(\mathcal{C}_x, \mathcal{C}_{new})$;

---

## C  PSEUDO-CODE FOR APPROXIMATE SEMANTIC SHIFT COMPUTATION

The pseudo-code for approximate Semantic Shift computation is depicted in Algorithm 3.

---
**Algorithm 3:** Approximate Semantic Shift Computation

---
**Input:** Larger cluster $\mathcal{C}_x$ and smaller cluster $\mathcal{C}_y$;
**Output:** Approximate Semantic Shift $\Delta \tilde{F}_{sem}(\mathcal{C}_x, \mathcal{C}_{new})$;
1  $\boldsymbol{A}_x$ and $\boldsymbol{A}_y \leftarrow$ Construct two matrices from $\mathcal{C}_x$ and $\mathcal{C}_y$;
2  $\sigma_{max} \leftarrow$ Compute the maximum singular value from $\boldsymbol{A}_x$;
3  Compute $\Delta \tilde{F}_{sem}(\mathcal{C}_x, \mathcal{C}_{new})$ using Eq. (8);
4  **return** $\Delta \tilde{F}_{sem}(\mathcal{C}_x, \mathcal{C}_{new})$;

---

## D  DATASET DETAILS

In the experiments, we employ five publicly available, real-world datasets for performance evaluation. Below are the dataset details:

- **AG-News:** This dataset comprises over 1 million news articles from more than 2,000 sources (Zhang et al., 2015), commonly used for clustering, classification, ranking, and search. Each article is categorized under one of four labels: **Business**, **Sci/Tech**, **Sports**, and **World**.
- **AAPD:** This Arxiv Academic Paper Dataset (AAPD) contains 55,840 abstracts and subjects of computer science papers developed for multi-label classification tasks (Yang et al., 2018).

Table D2: Description for the categories/labels in FakeNewsCorpus.

| Label | Description |
|---|---|
| Bias | Sources that come from a particular point of view and may rely on propaganda, decontextualized information, and opinions distorted as facts. |
| Clickbait | Sources that provide generally credible content but use exaggerated, misleading, or questionable headlines, social media descriptions, and/or images. |
| Conspiracy | Sources that are well-known promoters of kooky conspiracy theories. |
| Fake | Sources that entirely fabricate information, disseminate deceptive content, or grossly distort actual news reports. |
| Hate | Sources that actively promote racism, misogyny, homophobia, and other forms of discrimination. |
| JunkSci | Sources that promote pseudoscience, metaphysics, naturalistic fallacies, and other scientifically dubious claims. |
| Political | Sources that provide generally verifiable information in support of certain points of view or political orientations. |
| Reliable | Sources that circulate news and information in a manner consistent with traditional and ethical practices in journalism. |
| Satire | Sources that use humor, irony, exaggeration, ridicule, and false information to comment on current events. |
| Unreliable | Sources that may be reliable but whose contents require further verification. |

- **IMDB:** This dataset includes 50,000 movie reviews,[2] which is evenly split into 25,000 reviews for training and 25,000 for testing, used for binary sentiment classification (Maas et al., 2011).
- **Yelp:** This dataset contains millions of reviews, user data, and business attributes,[3] supporting tasks like sentiment analysis and business recommendations.
- **FakeNewsCorpus** This dataset was compiled by automatically scraping the list of domains from OpenSources (Pathak & Srihari, 2019). It contains 9,408,908 articles from 745 domains.[4] The corpus is primarily intended to train deep learning algorithms for fake news detection. We currently consider the news articles that are categorized based on their domains under ten labels: **Bias**, **Clickbait**, **Conspiracy**, **Fake**, **Hate**, **JunkSci**, **Political**, **Reliable**, **Satire**, and **Unreliable**. The descriptions of these ten labels are presented in Table D2.

**Experiment Environment.** All methods were written in Python 3.8. All experiments were conducted on a machine with Intel® Xeon® Platinum 8480C, 2.0 TB memory, and one NVIDIA H100, running on Ubuntu 20.04.

# E  EXTRA EXPERIMENTAL RESULTS

## E.1  MORE ANALYSIS ON HIERARCHICAL SEMANTIC STRUCTURE

In this section, we explore the differences between the hierarchical semantic structures identified by SAFARI in embedding spaces and the more intuitive hierarchies found in natural human language. In human language, semantics typically follow a logical hierarchy, progressing from specific, concrete entities to more abstract concepts, much like an ontology. However, in embedding spaces, this progression is not always intuitive. The distinction between "specific" and "abstract" depends more on the data and model than on human reasoning, often leading to groupings that diverge from what we would expect based on natural language understanding.

For example, as illustrated in Fig. 10, USA basketball teams are first grouped with USA football teams, and later, sports teams from various locations are merged, as shown in Fig. 11. This follows a logical hierarchical structure, from more specific categories to broader ones. Yet, as shown in Fig. E1 (at iteration 19,790), entities such as horse racing clubs, companies, and events (e.g., 'Jockey Club') are merged with famous racing horses. This merging of horse racing happens thousands of iterations after the merging of football and basketball teams in the USA. Following the ontology-like

---
[2]https://www.imdb.com/
[3]https://www.yelp.com/dataset
[4]https://github.com/several27/FakeNewsCorpus

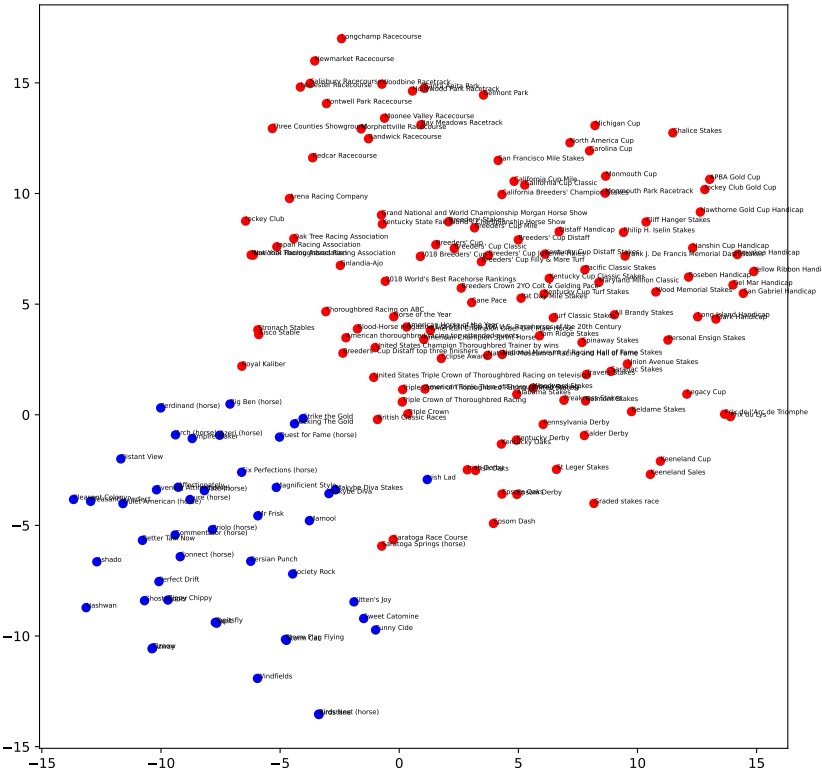

Figure E1: Famous racing horses merged with other horse racing entities.

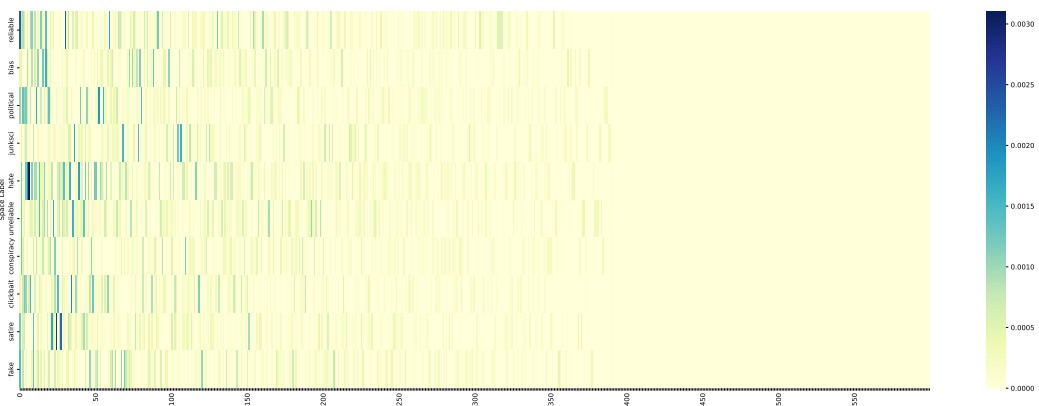

Figure E2: Explanation patterns of the Semantic Field Subspaces for the ten fake news categories.

progression, we would expect more abstract concepts. However, horse racing is not a more abstract concept compared with other sports.

These examples underscore that the hierarchical structures found in embedding spaces are shaped by the model's learning patterns, not by human logic. While some structures align with natural expectations, others can be surprising, revealing the complex relationship between the data and the model. This highlights the importance of careful interpretation when analyzing embeddings, as the resulting hierarchies may not always reflect conventional semantic reasoning.

### E.2 EXPLANATION PATTERNS FOR TEN FAKE NEWS CATEGORIES

Figure E2 presents the explanation patterns of the Semantic Field Subspaces constructed by SAFARI for the ten distinct fake news categories. Each subspace reveals different nearest entity types, showcasing the nuanced relationships between these categories. For instance, the subspaces highlight

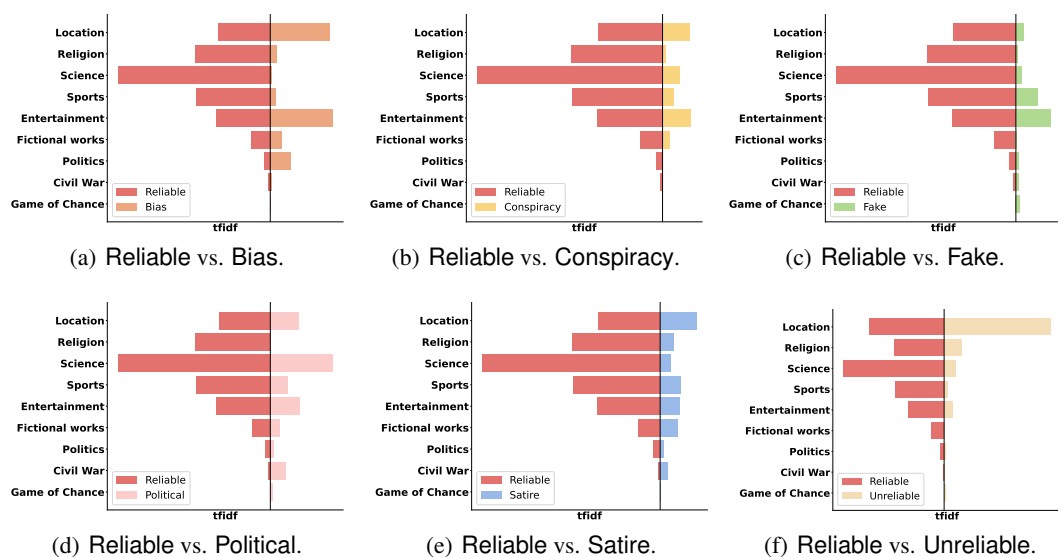

Figure E3: More comparisons between Reliable and other categories of fake news.

key differences in how content is structured across categories like Clickbait, Hate, and JunkSci, each emphasizing different entity types.

### E.3 MORE COMPARISON RESULTS BETWEEN RELIABLE AND OTHER CATEGORIES

Fig. E3 depicts more comparisons between Reliable and other categories of fake news. Before analyzing the results, we want to highlight that although this dataset is labeled as "fake news", the categories presented here are vague or not necessarily fake. These categories are not clearly distinct from each other; for instance, a news piece labeled as Fake could also fall under Bias or Unreliable simultaneously, making it unreasonable to strictly assign it to one category over another. Consequently, the analysis below may be based on an inherently uncertain foundation.

- **Bias**: The node types for Bias are similar to those for Reliable, particularly in areas like *Location* and *Entertainment* (see Fig. 3(a)). This is consistent with the dataset's description of biased content, which often involves propaganda or opinions presented as facts, rather than outright falsehoods. One notable difference is that Bias lacks prominence in node types like *Science*, *Religion*, and *Sports*, which appear more in Reliable news.
- **Conspiracy**: The node types for Conspiracy differ markedly from Reliable, with a notable emphasis on *Location* and *Entertainment* (see Fig. 3(b)). This aligns with the nature of conspiracy theories, which often center around speculative narratives involving covert operations or unverified claims. In contrast, Reliable news exhibits a more balanced spread across categories like *Science* and *Sports*, highlighting the factual and grounded nature of its content. The disparity between these node type patterns underscores the speculative and unverified themes prevalent in conspiracy content.
- **Fake**: The Fake category shows a scattered pattern in node types, with no strong emphasis on any particular theme (see Fig. 3(c)). This reflects the diversity and often random nature of fake news, which may cover a wide range of topics with little factual basis. Compared to Reliable news, which focuses on well-defined topics such as *Science*, *Religion*, and *Sports*, Fake news lacks consistency, mirroring the dataset's characterization of this category as containing misleading or fabricated stories.
- **Political**: As expected, the node type patterns for Political news are closely aligned with Reliable, with significant overlap in categories such as *Science*, *Location*, *Entertainment*, and *Sports* (see Fig. 3(d)). This similarity is consistent with the dataset's description of Political news as verifiable but presented with a specific viewpoint. The comparison with Reliable indicates that Political content, while biased toward certain ideologies or perspectives, does not stray far from reliable reporting in terms of the types of entities discussed.

- **Satire**: The Satire category shows high values in *Location*, *Entertainment*, and *Fictional works*, which contrasts sharply with the more factual node types (e.g., *Science* and *Religion*) found in Reliable news (see Fig. 3(e)). Satire often uses humor, exaggeration, and fictional scenarios to comment on real-world events, explaining the prominence of such node types. This divergence highlights the entertainment-driven and often fictional nature of satire, which deliberately distorts facts for comedic or critical purposes, unlike the more serious and factual content of Reliable news.
- **Unreliable**: The Unreliable category displays notable differences compared to Reliable news sources. Except for *Location*, the node types associated with Unreliable sources are scattered across diverse, less cohesive categories (see Fig. 3(f)). This suggests that Unlike Reliable news, which demonstrates a strong association with node types related to well-established factual domains like *Science*, *Religion*, and *Sports*, Unreliable sources exhibit a pattern of fragmentation. This reflects the unpredictable and often erratic nature of unreliable news content, where the underlying information may lack verification or coherence.

