# OpenReview forum: "A Single Swallow Does Not Make a Summer: Understanding Semantic Structures in Embedding Spaces"
_ICLR.cc/2025/Conference — Submitted to ICLR 2025_

### Official Review · Reviewer_T896 · 2024-10-18

**Soundness:** 1
**Presentation:** 1
**Contribution:** 2
**Rating:** 3
**Confidence:** 2

**Summary:**

**Title**: A Single Swallow Does Not Make a Summer: Understanding Semantic Structures in Embedding Spaces

**Summary**: The authors present a novel algorithm titled SAFARI, which is claimed to discover hierarchical semantic structures in embedding spaces.

**Strengths:**

The authors introduce a novel algorithm inspired by classical hierarchical clustering. Throughout their work, they attempt to provide deep motivations for their design choices, although these justifications often appear somewhat forced or artificial.

The study aims to address the interpretability of embedding spaces generated by models such as BERT and Word2Vec. This is one of the most crucial yet poorly understood components of deep learning architectures in Natural Language Processing (NLP).

The proposed algorithm and accompanying theory could prove valuable in various scenarios where practitioners seek to interpret a model's embeddings. This potential for practical application is a noteworthy aspect of the work.

**Weaknesses:**

### Major Comments

1. On line 160, the authors define the "interpretable semantics" of an embedding vector, $f_{int}(v)$. Their definition is based on the previously defined "close neighborhood" of said embedding vector, $\mathcal{N}(v)$. In particular, they define $f_{int}(v)$ as any subset of $\mathcal{N}(v)$. This implies that there are several "interpretable semantics" for each embedding vector. Furthermore, their notion of "interpretable semantics" is simply the intersection between the semantics of close neighborhood embeddings. It's unclear how this is "interpretable."

2. In lines 180-200:
   - The authors rightfully claim that $f_{sem}$ is not computable and introduce the concept of "semantic field subspace" to address this issue.
   - They motivate the choice to substitute a set of embedding vectors with a subspace of $\mathbb{R}^d$, presumably to make the "semantic field" computable.
   - However, their definition of "semantic field subspace" is still based on the non-computable $f_{sem}$, which seems to achieve very little.
   - In definition 5, they approximate the "subspace" with a finite set of vectors, which raises the question of why the "semantic field subspace" was introduced in the first place.

   This section feels like a convoluted and unconvincing justification for substituting $f_{sem}(v)$ with $v$. The argument would be more compelling without this section.

3. In line 197 (definition 5), the authors state that the subspace $\mathbb{S}$ can be approximated by a matrix $A$. This is unclear, as I am interpreting $\mathbb{S}$ as a simple subset of $\mathbb{R}^d$. Furthermore, the approximation $F_{sem}(\mathbb{S})\approx \Sigma,V^T$ is not well-explained. The authors should detail this approximation more clearly.

4. From what I can gather, each "Semantic Field Subspace" is simply a set of embeddings that the SAFARI algorithm found to be particularly semantically coherent. If this is the case, the name "Semantic Field Subspace" is somewhat confusing as it suggests a relation with fields and vector spaces that is not evident in the paper. I suggest the authors consider different nomenclature. I have similar concerns about the "Semantic Field" name.

5. On line 247, the authors use the threshold $\mu + 2\tau$ to detect new clusters. They should better motivate this choice. It might be more natural to use a hyperparameter to control how much semantic shift should be allowed in each cluster.

6. In section 5.2, it's unclear whether SAFARI accounts for the initial embedding training step (e.g., Word2Vec or BERT) in Figure 7. If not, the authors should mention this in the paper.

7. The authors use clusters generated by SAFARI for text classification and compare it to alternatives such as SVM, BERT, and KNN. However, from what I can gather from their code, it appears these algorithms are trained on the top-n entities rather than directly on the textual data (which would be natural for BERT). The authors should clarify the training procedure for the baselines, at least in the appendix.

8. The evaluation of SAFARI seems weak overall:
   - Section 5.2 tests for text classification, but SAFARI's purpose is not text classification.
   - Section 5.1 tests semantic field subspace alignment with BLINK entities at the dataset level, which seems less natural than using a single text classification dataset.
   - Section 5.4 presents only an anecdotal example showcasing SAFARI's ability to uncover hierarchical structures.
   - The absence of baselines makes it difficult to assess SAFARI's improvement over the state of the art.

   I suggest the authors consider a more comprehensive evaluation.

9. If SAFARI is indeed a clustering algorithm, shouldn't it be compared with other clustering algorithms rather than BERT or SVM?

### Minor Comments

1. On line 118, the authors mention the "deep learning" model $h:\mathcal{X}\rightarrow \mathcal{E}$, but don't use it meaningfully in the rest of the paper. It would be clearer to simply describe $\mathcal{E}$ as a set of embedding vectors, each representing a word in the vocabulary, possibly trained using a deep learning method.

2. On line 128, I believe the authors meant the power set of $\mathcal{M}$ (excluding the empty set) when writing $2^{|\mathcal{M}|}\setminus\emptyset$. It would be clearer to write $2^{\mathcal{M}}\setminus\emptyset$.

3. In lines 125-135, the authors' argument about words not being fully interpretable in isolation seems flawed. Their example of "Apple" as both a fruit and a tech company doesn't support their point, as $f_{sem}(\text{"apple"})$ could simply return $\{\text{"fruit"}, \text{"tech company"}\}$. The John Rupert Firth quote also seems irrelevant to their argument.

4. On line 154, the authors' method for characterizing the "close neighborhood" of an embedding seems arbitrary. Using k-NN with $k=3$ to exclude strict synonyms may exclude relevant embeddings if a word has fewer than 3 synonyms. A distance-based test might be more appropriate.

5. On line 167, the authors use the term "field" in a non-mathematical sense, which could be confusing. Consider changing the name to avoid this confusion.

6. In Algorithm 1, line 5 (line 224), the authors write $d_{sem}(\mathcal{C}_i,\mathcal{C}_j)$, but $\mathcal{C}_i$ and $\mathcal{C}_j$ are sets of vectors while $d_{sem}$ is a distance function on vectors. The distance function should be specified more clearly.

7. The authors use $\tau$ to denote standard deviation. It would be more conventional to use $\sigma$ instead.

8. On line 389, the BERT reference points to the wrong paper.

**Questions:**

My primary concerns center on the paper's narrative and experimental validation. I am open to reconsidering my assessment of the paper in a future revision, particularly if the following points are addressed:

1. Experimental Comparison:
   - Provide experiments that demonstrate superior clustering capabilities compared to other clustering approaches.
   - Note: Algorithmically generated datasets would be acceptable to keep the scale of experiments manageable.

2. Narrative Refinement:
   - Consider shifting the focus from "an interpretability tool" to a "clustering algorithm for interpreting embeddings."
   - Improve the nomenclature, or at least provide a clear rationale for the current terminology.

I hope the authors find this review constructive. Despite my current reservations about the paper, I believe there is potential in this work and look forward to seeing how it evolves in the review process.

---

> ### Author Response · Authors · 2024-11-25
>
> We sincerely thank the reviewer for their time, effort, and thoughtful feedback on our work. The detailed comments and constructive suggestions have provided valuable insights, helping us to clarify and improve the presentation of our contributions. Below, we address each point raised to ensure our responses are as clear and comprehensive as possible.
>
> Much of the confusion in the feedback appears to stem from a misunderstanding of the term Semantic Field Subspace. To ensure a clearer understanding, we recommend that the reviewer start with our response to this comment. Beginning with this response will provide essential context, addressing several misconceptions and clarifying the intended meaning.
>
> Marjor Comments
>
> > 4. From what I can gather, each "Semantic Field Subspace" is simply a set of embeddings that the SAFARI algorithm found to be particularly semantically coherent. If this is the case, the name "Semantic Field Subspace" is somewhat confusing as it suggests a relation with fields and vector spaces that is not evident in the paper. I suggest the authors consider different nomenclature. I have similar concerns about the "Semantic Field" name.
>
>
> * Thank you for bringing this to our attention. We appreciate your feedback and agree that we should have provided a more thorough explanation in the paper.
> *    "Field" in "Semantic Filed Subspace" and "Semantic Field" is not the mathmatical term "field" in math or physics. ["Semantic Field"](https://en.wikipedia.org/wiki/Semantic_field) is originally a term in linguistics, which refers to a lexical set of words grouped semantically, just like the set of embeddings that SAFARI found to be particularly semantically coherent as the reviewer mentioned.
> *    The term "Subspace" here is indeed used in the mathematical sense, consistent with the reviewer's understanding. However, SAFARI is not limited to discovering sets of semantically coherent embeddings. In Line 9 of the SAFARI Algorithm (Algorithm 1), we perform SVD on the aforementioned set of embeddings to construct a subspace (Algorithm 2) to compute Semantic Shift. The input for computing Semantic Shift is the subspace itself—specifically, the row space and singular values derived from SVD. Therefore, the subspace is a fundamental component of our algorithm.
> *    The Semantic Field Subspace is not merely a set of vectors but an actual subspace within the embedding vector space. This subspace generalizes the semantic meaning beyond the individual vectors, as evidenced by our experiments, which demonstrate that the subspace itself encapsulates the semantics effectively.
>
> > 1. On line 160, the authors define the "interpretable semantics" of an embedding vector $f_{int}(v)$. Their definition is based on the previously defined "close neighborhood" of said embedding vector, $\mathcal{N}(v)$. In particular, they define $f_{int}(v)$ as any subset of $\mathcal{N}(v)$. This implies that there are several "interpretable semantics" for each embedding vector. Furthermore, their notion of "interpretable semantics" is simply the intersection between the semantics of close neighborhood embeddings. It's unclear how this is "interpretable."
>
> * Firstly, we would like to clarify that $f_{int}(v)$ is not $\textit{any}$ subset of $\mathcal{N}(v)$. Selecting arbitrary elements from $\mathcal{N}(v)$ is unlikely to yield a clear meaning.
> * Regard of what is interpretable, the reviewer is correct in stating "there are several 'interpretable semantics' for each embedding vector." And we are unsure about which one should be chosen to 'interpret' the embedding vector.
> * We regard the process of refining meanings as interpretation. As intersections are performed, the semantics become increasingly refined and clearer, ultimately leading to unambiguous semantics. Of course, if we continue performing intersections, we would eventually reach an empty set.
> * Thus,  it is evident that that we cannot choose other $v$s randomly (or any subset of $\mathcal{N}(v)$). This is why we employ hierarchical clustering in the algorithm. It enables us to perform intersections with other semantically similar embedding vectors in a structured manner.
> *    An example of "Apple" is illustrated in Figure 1. Initially, there may be several interpretable semantics for "Apple," making its meaning ambiguous—or polysemiotic. As more neighbors are added (corresponding to the hierarchical clustering in our algorithm), the semantics become increasingly refined and clearer. Gradually, it becomes possible to determine whether this "Apple" refers to the fruit or the tech company. This progression from ambiguity to clarity is what we regarded as interpretation.
> *   In the revised version of the paper, we will polish this part in the main paper.

---

> > ### Author Response · Authors · 2024-11-25
> >
> > > 2. In lines 180-200:
> >     The authors rightfully claim that $f_{sem}$ is not computable and introduce the concept of "semantic field subspace" to address this issue.
> >     They motivate the choice to substitute a set of embedding vectors with a subspace of $\mathbb{R}^d$, presumably to make the "semantic field" computable.
> >     However, their definition of "semantic field subspace" is still based on the non-computable $f_{sem}$,which seems to achieve very little.
> >     In definition 5, they approximate the "subspace" with a finite set of vectors, which raises the question of why the "semantic field subspace" was introduced in the first place.
> >
> > * From a computational perspective, we acknowledge the reviewer's point that the definition of a Semantic Field Subspace may seem limited, as it is ultimately approximated using a set of vectors (a matrix).
> > * However, the Semantic Field Subspace is a crucial concept for understanding embedding spaces, which is the central goal of this paper. By treating singular vectors and singular values (obtained via SVD) as the basis and weights of subspaces, we interpret the semantic patterns underlying different regions and directions within the embedding space. These Semantic Field Subspaces represent intrinsic properties of the embedding space itself, independent of specific sets of vectors. As subspaces, they provide a framework for leveraging semantic patterns within a vector space model.
> > * Without this concept, we would merely extract semantic features from individual matrices (sets of vectors), which would not contribute to a deeper understanding of the embedding space.
> >
> > > 3. In line 197 (definition 5), the authors state that the subspace $\mathbb{S}$ an be approximated by a matrix $A$. This is unclear, as I am interpreting $\mathbb{S}$ as a simple subset of $\mathbb{R}^d$. Furthermore, the approximation $F_{sem}(\mathbb{S}) \approx \Sigma,V^\top$
> >  is not well-explained. The authors should detail this approximation more clearly.
> >  * Thank you for bringing this to our attention. We will make the necessary refinements.
> >
> > > 5. On line 247, the authors use the threshold $\mu +2 \tau$ to detect new clusters. They should better motivate this choice. It might be more natural to use a hyperparameter to control how much semantic shift should be allowed in each cluster.
> > * We appreciate your insights on this matter. To clarify, the threshold is not used to detect new clusters but rather works in conjunction with the sliding window to identify relatively large Semantic Shifts.
> >
> > * In hierarchical clustering, data points are iteratively merged based on their similarity. When considered globally, the Semantic Shifts are generally small enough to be negligible at the initial stage, which is not desirable. Therefore, a dynamic threshold is necessary rather than a fixed hyperparameter.
> >
> > > 6. In section 5.2, it's unclear whether SAFARI accounts for the initial embedding training step (e.g., Word2Vec or BERT) in Figure 7. If not, the authors should mention this in the paper.
> >
> > * We appreciate your feedback. The settings that are shared by all experiments are stated in Section 5.1, lines 355 to 364. We did not repeat these settings in each subsection for individual experiments due to page limitations.
> >
> > * To address this specific comment, SAFARI does not account for initial embedding training, and the training time in Figure 7 is not for initial embedding training as well. We will make the setting clearer.
> >
> >
> > > 7. The authors use clusters generated by SAFARI for text classification and compare it to alternatives such as SVM, BERT, and KNN. However, from what I can gather from their code, it appears these algorithms are trained on the top-n entities rather than directly on the textual data (which would be natural for BERT). The authors should clarify the training procedure for the baselines, at least in the appendix.
> >
> > * Thank you for pointing this out. We may not have explained the training process clearly enough. To clarify, the training is not focused on learning embeddings but is specifically aimed at the classification task.

---

> > > ### Author Response · Authors · 2024-11-25
> > >
> > > > 8. The evaluation of SAFARI seems weak overall
> > >
> > > > 8.1 Section 5.2 tests for text classification, but SAFARI's purpose is not text classification.
> > > * Yes. SAFARI's purpose is to discovery semantic patterns in embedding space, which we defined as Semantic Filed Subspace. If it is effective, it should be able to be used for classification. That is why we tested it with a classification task. We are not aware of specialized benchmarks tailored for the proposed method.
> > >
> > >
> > > > 8.2 Section 5.1 tests semantic field subspace alignment with BLINK entities at the dataset level, which seems less natural than using a single text classification dataset.
> > > * Section 5.1 was intented to show that Algorithm 1 can extract meaningful and distinct Semantic Field Subspaces. In Section 5.2, we conducted a text classification experiment.
> > > * We will consider expanding it to provide supplementary results on classification.
> > >
> > > >8.3 Section 5.4 presents only an anecdotal example showcasing SAFARI's ability to uncover hierarchical structures.
> > > * Thank you for your observation. Section 5.4 serves as a case study. However, since it addresses a novel task that we introduced, it is challenging to identify a hierarchically labeled dataset or an existing benchmark suitable for testing.
> > >
> > > > 8.4 The absence of baselines makes it difficult to assess SAFARI's improvement over the state of the art.
> > > > 9. If SAFARI is indeed a clustering algorithm, shouldn't it be compared with other clustering algorithms rather than BERT or SVM?
> > >
> > > * We would like to clarify that SAFARI is not designed for clustering. For further details, please refer to **The Clarification of Motivation** section in **To All Reviewers**.
> > > * Additionally, we believe that there is no established state-of-the-art for this specific task.
> > >
> > >
> > > Minor Comments
> > >
> > > * We sincerely thank you for your valuable feedback on the minor comments. We greatly appreciate your thoughtful suggestions, which will undoubtedly help us improve the quality and clarity of our work. We will carefully address each point and make the necessary revisions to ensure our paper is highly polished.
> > >
> > > Questions
> > >
> > > > 1. Experimental Comparison:
> > > >  Provide experiments that demonstrate superior clustering capabilities compared to other clustering approaches.
> > > >     Note: Algorithmically generated datasets would be acceptable to keep the scale of experiments manageable.
> > > > 2. Narrative Refinement:
> > > > Consider shifting the focus from "an interpretability tool" to a "clustering algorithm for interpreting embeddings."
> > >
> > > * We believe the reviewer now understands that both the technical contribution and the algorithm SAFARI do not focus that much on clustering.
> > >
> > > > Improve the nomenclature, or at least provide a clear rationale for the current terminology.
> > >
> > > * Thank you for pointing out the issue with the nomenclature.
> > > * We addressed this in our response for the 4th Major Comment. It is indeed an important issue that we had overlooked. We failed to sufficiently emphasize the linguistic term semantic field or adequately highlight the significance of the subspace, which may have easily misled readers into thinking SAFARI is a clustering algorithm.

---

> > > > ### Comment · Reviewer_T896 · 2024-11-26
> > > >
> > > > I have re-read the paper with the insights provided by the reviewers. In my initial review, I misunderstood certain concepts, which led to some incorrect statements. I apologize for the oversight.
> > > >
> > > > I still have a few concerns:
> > > >
> > > > 1) In Equation 6, it seems that any single vector from $\mathbb{S}$ will minimize the symmetric difference. For example, given any embedding $u \in \mathcal{E}$, if we set $\mathcal{C} = \{u\}$, the symmetric difference in Equation 6 simplifies to $|f_{sem}(u) - f_{sem}(u)|$, which equals 0. This does not appear to align with the authors' intentions. Could the authors clarify this?
> > > >
> > > > 2) The authors claim that the concept of a Semantic Field Subspace (SFS) enhances the interpretability of high-dimensional embeddings. While the SFS concept may indeed help better approximate a "semantic" distance, I still do not understand how it contributes to improving the interpretability of the embedding matrix.
> > > >
> > > > The following have already been stated. However, clarifying these issue will definitely raise the paper quality.
> > > >
> > > > 3) The approximation $F_{sem}(\mathbb{S}) \approx \Sigma_x, V_x^T$ requires stronger justification. Why should distances in the semantic space be approximated by the singular values and vectors of an SFS? While the intuition is somewhat clear, providing a more detailed motivation would significantly strengthen the argument.
> > > >
> > > > 4) In the experiments section, only one (Section 5.2) out of the four experiments includes a baseline and a best score for comparison. The other experiments (Sections 5.1, 5.4, and 5.5) offer only anecdotal evidence. I understand that evaluating this novel approach may pose challenges; however, this remains one of the paper's weakest points.

---

> > > > > ### Author Response · Authors · 2024-11-27
> > > > >
> > > > > > 1. In Equation 6, it seems that any single vector from $\mathbb{S}$ will minimize the symmetric difference. For example, given any embedding $\mu \in \mathcal{E}$, if we set $\mathcal{C}=\mu$, the symmetric difference in Equation 6 simplifies to $|f_{sem}(\mu)-f_{sem}(\mu)|$, which equals 0. This does not appear to align with the authors' intentions. Could the authors clarify this?
> > > > > * Thank you for highlighting this. Assuming $\mathcal{C} = \\{\mu\\}$, the Semantic Field Subspace is constructed with only one vector, $\mu$. And $|f_{sem}(\mu)-f_{sem}(\mu)|$ would end up with 0. But such vectors are are excluded.
> > > > >
> > > > > * Though we can not explicitly define the semantics of an embedding vector, we do specify one important constraint when constructing the Semantic Filed of it, that is excluding of synonyms ($k\text{-NNs}(v)$). This is specified in Line 149~157 and summariezed by the equation follows.
> > > > >
> > > > > * $\mathcal{N}(v) = \{v^\prime \mid f_{sem}(v^\prime) \cap f_{sem}(v) \neq \varnothing, v^\prime \in \mathcal{E} \} \setminus k\text{-NNs}(v).$
> > > > > * This exclusion constraint is currently positioned too far from Equation 6, which might lead to readers inadvertently overlooking its relevance. To address this, we will include a reminder or reference to the constraint closer to Equation 6 to ensure it is not missed.
> > > > > * We also gives an example to illustrate this in Fig. 2, the vectors of ‘Coca-Cola’ and ‘Coke’ closest neighboors ($k\text{-NNs}(v)$), which are nearly identical but redundant. Such nearest neighboors should not be used to construct a Semantic Field Subspace.
> > > > > * Interestingly, in practice, the presence of identical/nearly identical vectors is not a significant issue as long as additional more qualified vectors are included. In SVD, these identical vectors effectively function as a single vector. Though this may increase computational cost, depending on the implementations.
> > > > > > 2. The authors claim that the concept of a Semantic Field Subspace (SFS) enhances the interpretability of high-dimensional embeddings. While the SFS concept may indeed help better approximate a "semantic" distance, I still do not understand how it contributes to improving the interpretability of the embedding matrix.
> > > > > * We acknowledge that we may not have been sufficiently clear. The Semantic Field Subspace (SFS) does not contribute to better approximating semantic distances or enhancing any computational processes. Instead, the need for SFS stems from our primary motivation: to understand and interpret the embedding space, rather than focusing on specific embedding vectors or matrices.
> > > > >
> > > > > * In our study, our primary objective is to understand the structure of the embedding space, which is inherently challenging to ascertain directly. Therefore, we have approached this by characterizing the structures within the embedding space through the vectors it contains.
> > > > >
> > > > > * Ultimately, our goal is to develop Semantic Field Subspaces (SFSs), represented by bases and weights. The bases of each SFS are vectors/directions within the original space, allowing for decomposition and interaction between SFSs. Additionally, we can compare a set of SFSs of interest with the original space to analyze the distribution of specific semantics within the original embedding space. The SFS model facilitates this process, making it largely independent of the original sets of vectors.
> > > > >
> > > > > > 3. The approximation $F_{sem}(\mathbb{S}) \approx \Sigma,V^\top$ requires stronger justification. Why should distances in the semantic space be approximated by the singular values and vectors of an SFS? While the intuition is somewhat clear, providing a more detailed motivation would significantly strengthen the argument.
> > > > > * We are considering the possibility of not to use the notations "$F_{sem}(\mathbb{S}) \approx \Sigma,V^\top$" and "$\mathbb{S} \approx A$" and instead expressing these concepts in words. Our intention is to make the definitions more concise using mathematical symbols. However, we recognize that the approximation signs may be confusing to readers.
> > > > > * $\Sigma,V^\top$ are not intended to approximate distances within the space. Instead, they are used to represent a subspace through bases and weights, specifically the weighted basis of the subspace we constructed. This corresponds to the row space of $A$, if that terminology is preferred.
> > > > > * The expression $F_{sem}(\mathbb{S}) \approx \Sigma,V^\top$ indicates an approximation because, ideally, if we could enumerate all vectors within the desired subspace, this approximation would become an equality, i.e., $F_{sem}(\mathbb{S}) = \Sigma,V^\top$. This would signify that we have obtained the weighted basis of this subspace, allowing all vectors within it to be represented by these bases.
> > > > > * However, this ideal scenario is not practical, as we can only utilize a finite number of vectors to construct the subspace. This limitation is what we mean by the expression $\mathbb{S} \approx A$.

---

> > > > > > ### Author Response · Authors · 2024-11-27
> > > > > >
> > > > > > > 4. In the experiments section, only one (Section 5.2) out of the four experiments includes a baseline and a best score for comparison. The other experiments (Sections 5.1, 5.4, and 5.5) offer only anecdotal evidence. I understand that evaluating this novel approach may pose challenges; however, this remains one of the paper's weakest points.
> > > > > > * Thank you for pointing out this issue.
> > > > > > * We are currently incorporating the results of different embedding models (RoBERTa, Electra, XLNet and MiniLM) in Section 5.2 to strengthen the experimental analysis. This will allow us to compare how SAFARI operates within the embedding spaces of various models, providing a clearer understanding of its performance and adaptability across different contexts. By evaluating SAFARI in diverse embedding environments, we aim to demonstrate its robustness and versatility.
> > > > > > * Additionally, we are considering an experiment to demonstrate the utility of the SFS by integrating it with Support Vector Machines (SVM). Since an SVM inherently determines a hyperplane/decision boundary, gaining a better understanding of the embedding space's structure could enhance its effectiveness, such as by excluding certain subspaces from consideration. One potential approach to achieve this is through active learning, although we feel this might constitute an entirely separate study.
> > > > > > * Currently, we are still exploring other potential tasks that would be suitable for inclusion in this work, but we welcome suggestions for any other interesting datasets or tasks that would be valuable for evaluation. If the reviewer could provide any recommendations, it would be greatly appreciated and helpful in enhancing our study.

---

> > > > > > > ### Comment · Reviewer_T896 · 2024-11-27
> > > > > > >
> > > > > > > I do not think I have expressed my first concern clearly. The author write (from line 187)
> > > > > > >
> > > > > > > > Let $\mathbb{S}$ be a subspace of $\mathbb{R}^d$. The semantics of $\mathbb{S}$ is defined as :
> > > > > > > > $$F_{sem}(\mathbb{S}) = \bigcup_{v \in \mathcal{C^*}} f_{sem}(v)$$
> > > > > > > > where $\mathcal{C}^*$ is the set of emebdding vectors in $\mathbb{S}$ that minimizes the following symmetric difference:
> > > > > > > > $$\mathcal{C}^* = argmin_{\mathcal{C}\subset{\mathbb{S}}}\\{|\bigcup_{v'\in\mathcal{C}}f_{sem}(v') - \bigcap_{v'\in\mathcal{C}}f_{sem}(v')|\\}$$
> > > > > > >
> > > > > > > I believe that the idea that the authors want to convey is that $F_{sem}$ can retrieve the semantics for any subspace of $\mathbb{R}^d$. Is this correct?
> > > > > > >
> > > > > > >
> > > > > > > Now, let $\mathbb{A}$ be any vector subspace of $\mathbb{R}^d$. We know that the zero-vector $\vec{0}$ belongs to $\mathbb{A}$, by definition of vector space.
> > > > > > >
> > > > > > > Let us compute the semantics of $\mathbb{A}$ according to the author definition. We have that
> > > > > > >
> > > > > > > $$F_{sem}(\mathbb{A})=\bigcup_{v \in \mathcal{C}^*} f_{sem}(v)$$
> > > > > > >
> > > > > > > Now, to compute $\mathcal{C}^*$, we need to minimize the symmetric difference. The minimum value that the symmetric difference can take is $0$. So, if we find a set $\mathcal{C}$ such that the symmetric difference is $0$, then we have found a set $\mathcal{C}^*$.
> > > > > > >
> > > > > > > So, let us choose $\mathcal{C} = \\{\vec{0}\\}$. Note that $\\{\vec{0}\\} \subset \mathbb{A}$ for any non-zero vector space. Now, we have that:
> > > > > > >
> > > > > > > $$\\{|\bigcup_{v'\in\mathcal{\\{\vec{0}\\}}}f_{sem}(v') - \bigcap_{v'\in\\{\vec{0}\\}}f_{sem}(v')|\\} = \\{|f_{sem}(\vec{0}) - f_{sem}(\vec{0})|\\} = 0$$
> > > > > > >
> > > > > > > So, it appears that $\\{\vec{0}\\}$ is a valid candidate as $\mathcal{C}^*$. Which, we can use to compute the semantics of $\mathbb{A}$ as:
> > > > > > >
> > > > > > > $$\bigcup_{v \in \\{\vec{0}\\}} f_{sem}(v) = f_{sem}(\vec{0}) $$
> > > > > > >
> > > > > > > Now, we have not made any significant assumption on $\mathbb{A}$ (apart being a subspace of $\mathbb{R}^d$ and not being $\\{\vec{0}\\}$), so we have the following:
> > > > > > >
> > > > > > > $$\forall \mathbb{A} vector subspace of  \mathbb{R}^d : F_{sem}(\mathbb{A}) = f_{sem}(\vec{0})$$
> > > > > > >
> > > > > > > This, of course, would mean that all subspace have the same semantics according to $F_{sem}$, which I do not think it is the intention of the authors.
> > > > > > >
> > > > > > > Furthermore, the fact that there could be multiple $\mathcal{C^*}$ for any vector subspace of $\mathbb{R}^d$ is somewhat concerning, as it would mean that each vector subspace could have multiple semantics.
> > > > > > >
> > > > > > > I may be wrong with this reasoning, I hope the authors can clarify this point.

---

> ### Author Response · Authors · 2024-11-29
>
> * Before addressing specific comments, we would like to clarify a potential misunderstanding related to mathematical notation that may have arisen due to differing conventions or our presentation. It appears that the reviewer has interpreted several definitions or terms in our paper with the prefix "**any**", which was not our intention.
> * Specifically, if the symbol $\forall$ does not appear in a definition or term, it should not be interpreted with the prefix "**any**" or "**for all**". For instance, **$\mathbb{S}$ is not intended to represent "**all**" possible subspace of $\mathbb{R}^d$ without constraints, and $\mathcal{C}$ is not intended to denote "**all**" possible set of vectors.** Misinterpretations of this nature could lead to unintended understandings of the work.
> > I do not think I have expressed my first concern clearly. The author write (from line 187).
> > > Let $\mathbb{S}$ be a subspace of $\mathbb{R}^d$. The semnatics of $\mathbb{S}$ is defined as:
> > > $F_{sem}(\mathbb{S}) =  \bigcap_{v \in \mathcal{C}^*} f_{sem}({v})$
> > where $\mathcal{C}^*$ is the set of embedding vectors in $\mathbb{S}$ that that minimizes the following symmetric difference:
> > $\mathcal{C}^* = argmin_{\mathbb{C} \subset \mathbb{S}} {\bigcup_{v' \in \Set{C}} f_{sem}(v') - \bigcap_{v' \in \Set{C}} f_{sem}(v')}.$
> >
> > I believe that the idea that the authors want to convey is that $F_{sem}$ can retrieve the semantics for any subspace of $\mathbb{R}^d$. Is that correct?
> * No. $F_{sem}$ is not intended to retrieve the semnatics for **any** subspace of $\mathbb{R}^d$. The process begins with the vector set $\mathcal{C}$, which is not arbitrary. $\mathcal{C}$ is first introduced as an input of $F_{sem}()$, the function used to construct Semantic Field, as extended from Definition 2. We would like to clarify that $\mathcal{C}$ is derived from the neighbor set of a certain vector $v$.
> * Additionally, the vector $v$ we mention in Section 3 is not randomly chosen within the space. Our discussion pertains to an embedding space defined as a vector space to utilize a broader array of mathematical tools. However, this does not imply that all concepts applicable to vector spaces are valid here, as embedding spaces are not fully understood. An obvious difference is that an embedding space is not that "dense", meaning not every vector within them is meaningful for analysis. Randomly selecting a vector from this space is likely to yield one lacking significance.
> * This also further limits the elements of $\mathcal{C}$ to obtain $\mathcal{C}^*$. In our study, only embedding vectors that can be mapped into entities are considered since it is an embedding space of an Entity Linking model. Therefore, the algorithm we provided, in practice, is more akin to searching than optimization.
> * To summarize, $F_{sem}$ is not supposed to retrieve the semnatics for **any** subspace of $\mathbb{R}^d$. The domain of this function is limited, because $\mathbb{S}$ does not mean a subspace without any constraints.
> > So, let us choose $\mathcal{C} = \\{\vec{0}\\}$.
> * As a result, any set containing only a single vector (e.g., $\\{\vec{0}\\}$) cannot be considered a valid $\mathcal{C}$. Moreover, the vector $\vec{0}$ has no practical significance in real-world applications, as it does not correspond to any meaningful entity. Consequently, $\vec{0}$ is not regarded as a valid $v$ in this study.
> * We hope these clarifications contribute to a clearer understanding of our intentions and approach. We are grateful for the reviewer's feedback and welcome any additional questions or suggestions.

---

> > ### Comment · Reviewer_T896 · 2024-11-29
> >
> > Thank you for the clarifications. I appreciate the authors taking the time to address my concerns.
> >
> > However, **Definition 4** states, "Let $\mathbb{S}$ be a subspace of $\mathbb{R}^d$," and **Equation 6** writes, "$\\text{argmin}_{\mathcal{C} \subset \mathbb{S}} |...|$." Based on a classical mathematical interpretation, these statements naturally lead to my original conclusion. Additionally, the authors' comments suggest that, in practice, $\mathbb{S}$ might simply refer to a subset of all the embeddings.
> >
> > To avoid taking up more of the authors' time, I have outlined my final considerations below. I hope these comments will be helpful in the authors' future endeavors.
> >
> > The authors present an interesting technique. However, the manuscript is poorly written. The intuitions behind the algorithm are inadequately discussed, and the mathematical notation appears inconsistent with conventional standards. These issues create confusion regarding the authors' theoretical intentions, making it difficult to assess the novelty of the work from a review perspective. Moreover, the experimental section fails to provide convincing evidence that the proposed method improves the understanding of embeddings.
> >
> > To bring this work to a publishable state, the authors should undertake a thorough revision of the manuscript, with particular attention to clarity, mathematical notation, and explanatory detail.
> >
> > For these reasons, I maintain my original scores.

---

> > > ### Author Response · Authors · 2024-12-01
> > >
> > > We appreciate the reviewer’s comments and the time dedicated to this work, which are invaluable for its improvement. We agree that the intuitions behind the algorithm require a more thorough discussion, and we recognize that our explanation relied on some implicit assumptions about the embedding space, which may have made the theoretical intentions harder to follow. To address these concerns, we will provide a clearer explanation and conduct additional experiments to strengthen the contributions of this work.

---

### Official Review · Reviewer_WKBX · 2024-10-27

**Soundness:** 3
**Presentation:** 3
**Contribution:** 2
**Rating:** 5
**Confidence:** 3

**Summary:**

This paper is well-structured and introduces a novel approach to understanding the semantic structure of embedding spaces. It presents the SAFARI algorithm, which aims to uncover hierarchical semantic structures within high-dimensional embedding spaces using a method based on hierarchical clustering and Semantic Shift analysis. The concept and motivation are both relevant and timely, given the increasing need to interpret embeddings in NLP. The technical aspects are adequately explained, though a few areas could benefit from additional clarification to enhance accessibility for a broader audience. Also, the authors evaluate the proposed model from diverse aspects, including token classification accuracy, computational complexity and visualization. This diverse experimental validation is interesting and throughout.

In conclusion, this paper presents a relatively new contribution to embedding space interpretability. Its main strengths lie in the innovation of the SAFARI algorithm and the diverse experimental validation. With further clarifications and added discussions on the limitations, this paper could fairly impact interpretability research in embedding models.

**Strengths:**

The introduction of Semantic Field Subspaces and the SAFARI algorithm is an interesting new approach. The approach is innovative and offers a structured way to interpret embedding spaces, potentially broadening the scope for further research in embedding interpretability.

The paper provides a comprehensive evaluation of SAFARI on multiple datasets, showcasing its performance in terms of accuracy, computational efficiency, and interpretability. This diverse evaluation adds credibility to the proposed method and its applicability across various real-world tasks.

The paper is well-written and well-organized. No English grammar errors are detected. The graphs and figures are presented with enough details. The proposed methods is well described.

**Weaknesses:**

Given all the merits mentioned in above, there is one relatively major defects: There is a lack of comparison, which is a common defects shared by many many papers. To this work, especially:

1. There is a lack of baseline models: The authors compared the classification accuracy and training time of the proposed model against SVM, KNN, Random Forest and BERT on the AG-News dataset. The only notable baseline models is the BERT model. All others, SVM, KNN and random forest, cannot be regarded as formal baselines. You need to find more published models to serve as baselines. Now that you adapt BERT, why not further include RoBERTa, Electra and XLNet? More published baseline models are required to make the comparison more convincing.

2. There is a lack of comparison scenario: The authors only conducted classification accuracy and training time comparison on the AG-News dataset. I think that is not enough. I am not familiar with this dataset, so I think the authors may provide a better background introduction to benchmark dataset used. Also, the authors may consider using more than one dataset for accuracy and training time comparison. In addition, beyond accuracy, how about F-1 score, precision, and recall comparison. Accuracy is sometimes not enough to fully evaluate the performance of a model.

**Questions:**

The authors need to find more published models to serve as baselines. Now that you adapt BERT, why not further include RoBERTa, Electra and XLNet?
 Why not conduct more comparisons on datasets other than AG-News?
 Why not use F-1 score, precision and recall other than classification accuracy?

---

> ### Author Response · Authors · 2024-11-25
>
> > Given all the merits mentioned in above, there is one relatively major defects: There is a lack of comparison, which is a common defects shared by many many papers. To this work, especially:
> There is a lack of baseline models: The authors compared the classification accuracy and training time of the proposed model against SVM, KNN, Random Forest and BERT on the AG-News dataset. The only notable baseline models is the BERT model. All others, SVM, KNN and random forest, cannot be regarded as formal baselines. You need to find more published models to serve as baselines. Now that you adapt BERT, why not further include RoBERTa, Electra and XLNet? More published baseline models are required to make the comparison more convincing.
>
> * We appreciate the suggestion and agree that adding more comparisons could be beneficial. Initially, our focus was on comparing with non-deep learning models due to their explainability.
> * BERT was included primarily to illustrate that this is not a trivial classification task. However, we acknowledge that incorporating additional deep learning models, such as RoBERTa, Electra, and XLNet, could enhance the persuasiveness of our results. We will update this section accordingly.
>
> > There is a lack of comparison scenario: The authors only conducted classification accuracy and training time comparison on the AG-News dataset. I think that is not enough. I am not familiar with this dataset, so I think the authors may provide a better background introduction to benchmark dataset used. Also, the authors may consider using more than one dataset for accuracy and training time comparison. In addition, beyond accuracy, how about F-1 score, precision, and recall comparison. Accuracy is sometimes not enough to fully evaluate the performance of a model.
>
> * AG-News is a widely used text classification dataset (Madaan et al. 2024; Wei et al. 2021; Zaheer et al. 2020). We appreciate your suggestion and agree that including more background information would be beneficial.
>
> **Reference**
>
> * Madaan, A., Tandon, N., Gupta, P., Hallinan, S., Gao, L., Wiegreffe, S., Alon, U., Dziri, N., Prabhumoye, S., Yang, Y. and Gupta, S., 2024. *Self-refine: Iterative refinement with self-feedback.* Advances in Neural Information Processing Systems, 36.
> *  Wei, J., Bosma, M., Zhao, V.Y., Guu, K., Yu, A.W., Lester, B., Du, N., Dai, A.M. and Le, Q.V., 2021. *Finetuned language models are zero-shot learners.* arXiv preprint arXiv:2109.01652.
> * Zaheer, M., Guruganesh, G., Dubey, K.A., Ainslie, J., Alberti, C., Ontanon, S., Pham, P., Ravula, A., Wang, Q., Yang, L. and Ahmed, A., 2020. *Big bird: Transformers for longer sequences.* Advances in neural information processing systems, 33, pp.17283-17297.

---

### Official Review · Reviewer_RvcZ · 2024-10-29

**Soundness:** 2
**Presentation:** 2
**Contribution:** 1
**Rating:** 3
**Confidence:** 3

**Summary:**

The paper proposes a method to understand embedding spaces through hierarachical clustering. The authors define semantic fields, semantic field subspaces, and semantic shifts. In the quest of quantifying semantic shifts, the paper introduces a method based on singular values. Then, one of the paper's key contributions is a method to approximate the semantic shift without the need for SVD, getting speed-ups of 15-30x.

**Strengths:**

- an interesting method to understand embeddings
- the paper is very clear on the methodology part, building on a solid mathematical foundation

**Weaknesses:**

- although the paper starts off with a generic motivation, it remains unclear how the proposed approach would tackle, e.g., mentioned problem of polysemiotic words
- some assumptions seem to be made without justification. Unless I'm missing something, hierarchical clustering only ever assigns a point to a single cluster. This does not play well with the key motivation that words are polysemous and meaning determined by their use in context.
- the experiment section is not very clear about what embeddings are used.
- the evaluation seems weak (details below).
- limited to text embeddings

My initial recommendation is to reject the paper. In short, although the method is interesting, the paper is not clear enough in connecting the proposed method to the motivation and the method seems to have major limitations that are either not addressed or at least unclear in the current form of the paper. Claims are vague (analyzing embedding spaces) and the experimental part is not very convincing as it does not fully capture the analysis/interpretability aspect laid out in the motivation.

For improving the paper (though going beyond the scope of the rebuttal), it would be interesting to see the method applied to contextualized word embeddings, e.g., of decoder-only language models, and maybe non-language embedding models, e.g., on images, graphs etc. This would support the generality that the authors suggest in the beginning of the paper.

**Questions:**

Q1: How do you disambiugate polysemous words in the clustering? It seems that the method only assigns a word to a single cluster?

Q2: What embeddings are used for the experiments? Are they just tf-idf vectors? word2vec? contextualized embeddings from a language model?

---

> ### Author Response · Authors · 2024-11-25
>
> Weakness
> > 1. although the paper starts off with a generic motivation, it remains unclear how the proposed approach would tackle, e.g., mentioned problem of polysemiotic words.
>
> * This aspect is handled through Entity Linking when obtaining the embedding. We would like to emphasize that the primary motivation of this paper is not to address the issue of polysemiotic words.
> * For further details, please refer to **The Clarification of Motivation** in **To All Reviewers**.
>
> > 2. some assumptions seem to be made without justification. Unless I'm missing something, hierarchical clustering only ever assigns a point to a single cluster. This does not play well with the key motivation that words are polysemous and meaning determined by their use in context.
>
> * Hierarchical clustering is not used to address the polysemiotic words problem.
> * Please refer to **The Clarification of Motivation** in **To All Reviewers**.
>
> > 3. the experiment section is not very clear about what embeddings are used.
>
> * Plese refer to **Unclear Experiment Setting** in **To All Reviewers**.
>
> > 4. limited to text embeddings
>
> * Thank you for your insightful suggestion. Testing the method on other types of embeddings, such as image or graph embeddings, is indeed an intriguing idea. We will include this as a potential direction for future work.
>
> > Q1: How do you disambiugate polysemous words in the clustering? It seems that the method only assigns a word to a single cluster?
>
> * This part is done by Entity Linking when we get the embedding. The clustering algorithm is not for this purpose. Please refer to **The Clarification of Motivation** in **To All Reviewers**.
>
> > Q2: What embeddings are used for the experiments? Are they just tf-idf vectors? word2vec? contextualized embeddings from a language model?
>
> * They are entity embeddings from BLINK.
> * Please refer to **Unclear Experiment Setting** in **To All Reviewers** for more details.

---

> > ### Comment · Reviewer_RvcZ · 2024-11-26
> >
> > I have read the author response, other reviews, and looked again into the paper. Thank you for the clarifications. I have raised my ratings accordingly.

---

### Official Review · Reviewer_qNrg · 2024-10-30

**Soundness:** 1
**Presentation:** 2
**Contribution:** 2
**Rating:** 3
**Confidence:** 4

**Summary:**

The paper presents a new approach SAFARI, which is a novel algorithm for Semantic Field subspace determInation. It uses hierarchical clustering to uncover semantic structures present in a set of data representations. The paper claims to improve efficiency of the clustering process by efficiently quantifying the semantic drift during clustering. The paper presents several results on text classification datasets and analysis experiments to evaluate their method.

**Strengths:**

1. The paper tackles an important problem of automatically uncovering latent semantic concepts within a set of data representations.
2. The paper uses several visual illustrations (Fig. 1-3) that help the reader understand better.

**Weaknesses:**

1. The soundness and novelty of the paper are not substantial. Specifically, the paper builds on a simple agglomerative clustering algorithm and modifies the cluster merging logic using semantic drift.
2. The initial part of the paper is quite difficult to understand. This is mainly due to the inconsistent use of mathematical notations. Some of the claims in the Theorem proofs are incorrect. Please find my concerns below:

Definition 5 is confusing. What is matrix $\mathbf{A}$? It wasn’t defined before. What does it mean that $\mathbf{A}$ can be approximated by $\mathbb{S}$?

What does the notation $F_{sem}(\mathbb{S}) ≈ \Sigma, V^\top$ mean? How can a matrix be approximated using two matrices?

I’m unable to follow the intuition behind Eq. 7. What does the difference signify? Earlier $F_{sem}(\mathbb{S})$ was defined as a tuple of two matrices $\Sigma, V^\top$.

Line 265: In the example, it is unclear to me how Fig. 4 is formed after just 3 iterations. In Algorithm 1, Line 5, the merging happens in a cluster-wise fashion, there are 8 merges in Fig. 4.


$\sigma_i([A_x|O]) = \sigma_i(A_x)$: this isn’t true as there are additional 0 eigenvalues for the matrix on the left.

How is Equation 8 proved after Theorem 2? The approximation is an upper bound on the original value.

$r, r_1, r_2$ are not defined in Eq. 13.

3. The presentation of the experimental section needs to be improved. The section should start by stating the overall goal of the empirical results and analysis experiments. Currently, the reader is confused about several setup issues. Please find my concerns below:

The experimental setup in Section 5.2 isn’t clear to me. Are you using labels during training? If yes, then doesn’t the reported results significantly underperform simple neural network classification baselines on top of BERT?

If the paper is posed as a new hierarchical clustering method, it should present results of dendogram accuracy on clustering datasets.

**Questions:**

Please respond to the questions in the Weaknesses section.

---

> ### Author Response · Authors · 2024-11-25
>
> Before addressing the specific queries, we would like to gently remind Reviewer qNrg of a possible misunderstanding regarding the distinction between singular values/vectors and eigenvalues/eigenvectors. We understand this may be a minor oversight, but mistaking singular values/vectors for eigenvalues/eigenvectors could indeed create challenges in comprehending the remainder of our work. We believe this may have influenced several of the comments provided. We hope this clarification proves helpful as we proceed with addressing the individual queries.
>
> > Line 265: In the example, it is unclear to me how Fig. 4 is formed after just 3 iterations. In Algorithm 1, Line 5, the merging happens in a cluster-wise fashion, there are 8 merges in Fig. 4.
>
> * Thank you for your comment regarding Figure 4. We would like to clarify that this figure serves as a *toy example* aimed at providing conceptual illustration. It is not intended to represent a case study and is not meant to correspond to the exact iterations of the algorithm.
>
> > $\sigma_i([A_x | O]) = \sigma_i(A_x)$: this isn’t true as there are additional 0 eigenvalues for the matrix on the left.
>
>
> * We would like to respectfully remind the reviewer that the terms in question are singular values, usually represented by $\sigma$, instead of eigenvalues, conventionally represented by $\lambda$. Zero singular values are typically ignored in the context of matrix perturbation and many other practical applications because they do not contribute to the structure of the matrix.
>
> > How is Equation 8 proved after Theorem 2? The approximation is an upper bound on the original value.
>
>
> * We would like to kindly remind the reviewer that on the left side of Equation 8, it is $\tilde{F}_{sem}(c_x,c_{new})$ rather than $F_{sem}(c_x,c_{new})$.
> * Furthermore, using bounds as an approximation is a common, practical, and effective practice in matrix perturbation theory. This approach forms the foundation of some of the most fundamental and well-known theorems in the field, such as the Davis-Kahan Theorem and Weyl's Theorem.
>
> > The presentation of the experimental section needs to be improved. The section should start by stating the overall goal of the empirical results and analysis experiments. Currently, the reader is confused about several setup issues.
>
> * Please refer to **Unclear Experiment Setting** in **To All Reviewers**.
>
> > $r$,$r_1$,$r_2$ are not defined in Eq. 13.
>
> * We would like to clarify that the $r$, with or without subscripts, denotes the rank of a matrix. We believed this is evident given the context of a discussion closely related to singular values and vectors. However, we appreciate the reviewer's attentiveness, and we will make sure to explicitly define these terms to enhance clarity.
>
> > If the paper is posed as a new hierarchical clustering method, it should present results of dendogram accuracy on clustering datasets.
>
> * Thank you for your question. No, the paper is not.
> * For further details, please refer to **The Clarification of Motivation** in **To All Reviewers**.

---

### Official Review · Reviewer_HdVR · 2024-11-07

**Soundness:** 3
**Presentation:** 3
**Contribution:** 2
**Rating:** 5
**Confidence:** 3

**Summary:**

This paper proposes a method named SAFARI which constructs a cluster of semantic embeddings (I interpret as some meaningful word/phrase embeddings) from a set of such embeddings. The clustering algorithm is bottom-up hierarchical in nature, but employees a custom merge criteria based on their proposed semantic shift computation (Fig 4). The semantic shift between two clusters (hence two lists of embeddings) is calculated by first treating the list of embeddings as a matrix, and then finding their eigen-value and vectors, and finally taking the difference (eq 7). The authors noted that this process can be a bottleneck for the runtime because of the need of performing SVD, and proposed a approximation to this computation that seems to work well in the following experiments. Experimentally, the authors tested how strong their algorithm can separate embeddings from datasets coming from different domain, and compared with some baselines on classifying the embeddings of different textual classes.

**Strengths:**

The paper proposes an interesting angle to semantic embedding spaces by quantifying it as the semantic shift when as new embedding is introduced.  An efficient workaround to their semantic shift metric that requires SVD is introduced, and might be useful in other scenarios where a similar calculation is needed.

**Weaknesses:**

1. The experiment section needs some more details, It is missing details on how the baselines are adapted in the work, the hyperparameters chosen, and how the dataset for the experiment is constructed.
2. The intuition behind the semantic shift in eq (7) is lacking. It seems quite a big jump from an abstract definition of semantic fields to a concrete definition based on the differences of their eigen-values and eigen-vectors.

**Questions:**

1. The two weaknesses might be addressable via detailed experimental configurations and better intuition.
2. There are many equations in the paper, the ones that are not referenced later probably does not need a label.

---

> ### Author Response · Authors · 2024-11-25
>
> Weakness
> > W1. The experiment section needs some more details, It is missing details on how the baselines are adapted in the work, the hyperparameters chosen, and how the dataset for the experiment is constructed.
>
> *   We sincerely appreciate your insightful feedback regarding the experimental configurations.
> *   We have realized that one issue with the writing in the experiment section is that the shared experimental settings are only listed in Section 5.1. As a result, it may be easy for readers to overlook them when reviewing the subsections for the remaining experiments. Additionally, we acknowledge that we did not clearly specify the configurations of the baseline models, particularly regarding the training process, as also pointed out by ReviewerT896. This lack of clarity could potentially lead to misunderstandings and make the experiments appear less supportive.
> *   To address these concerns, we will provide a more detailed explanation of how the baseline models are utilized to ensure clarity.
>
> > W2. The intuition behind the semantic shift in eq (7) is lacking. It seems quite a big jump from an abstract definition of semantic fields to a concrete definition based on the differences of their eigen-values and eigen-vectors.
> *   Thank you for your query regarding the intuition behind Equation 7. In our approach, we approximate a Semantic Field Subspace using singular values (representing importance) and singular vectors (representing basis). When the semantics of a given subspace change, the basis of this subspace, or the importance of the basis, would change accordingly.
> *   For instance, if we randomly collect embeddings from the contexts of the word 'Apple' and use these embedding vectors to construct a subspace, two significant bases (or two sets of significant bases) may emerge: one capturing the meaning of Apple as a fruit and the other capturing its meaning as a technology company.
> * If we subsequently add more embeddings from contexts related to fruits, the importance (singular values) of the fruit-related basis will adjust, along with changes in the directions of the bases (singular vectors). Similarly, adding embeddings from tech-related contexts would impact the tech-related basis in the same manner.Hence, Equation 7 provides a straightforward method to quantify these changes.
>
> > Q2. There are many equations in the paper, the ones that are not referenced later probably does not need a label.
> * We appreciate your observation regarding the numerous equations in the paper. We acknowledge this and will carefully review the equations to remove labels from those that are not referenced. We believe that this revision will enhance the clarity and readability of the initial theoretical section, making it easier to follow for our readers.

---

### Author Response · Authors · 2024-11-25
**General Response To All Reviewers**

**1.  The Clarification of Motivation**
*  The motivation of the paper is to discovery semantic patterns (Semantic Field Subspace) in embedding space. Therefore, instead of extracting the common semantics from a set of vectors, the aim is to obtain geometric structures that carries semantic meanings in an embedding space.
* It is not focused on the problem of polysemiotic words. Our work **explicitly utilizes contextualized word embeddings**, which naturally separate embeddings for polysemiotic words based on their contexts.
* It is not a clusetring work. Our algorithm (SAFARI) not limited to discovering sets of semantically coherent embeddings.
    * In Line 9 of the SAFARI Algorithm (Algorithm 1), we perform SVD on the identified set of embeddings to construct a subspace (Algorithm 2) for computing Semantic Shift.
    * Our **technical contributions** lie in **the computation of Semantic Shift to further decide Semantic Field Subspace** and the **approximations used to accelerate the process**, enabling efficient separation of different hierarchical Semantic Field Subspaces.
    * This can be viewed as an analysis of the hierarchical clustering results. However, we do not claim the clustering itself to be a novel algorithm or assert any direct contributions to it.

**2. The intuition behind the semantic shift in Equation (7)**
* Regarding the intuition behind Equation 7, we approximate a Semantic Field Subspace with its singluar vectors (bases) and singular values (importance). When the semantics of a given subspace changes, its bases or the importance of these bases will change.
* For example, if we randomly collect embeddings from the contexts of the word 'Apple' and use these embedding vectors to construct a subspace, two significant bases (or two sets of siginificant bases) may emerge: one capturing the meaning of Apple as a fruit and the other capturing its meaning as a technology company.

**3. The linguistic meaning of Semantic Field**
* "Field" in "Semantic Filed Subspace" and "Semantic Field" is not the mathmatical term "field" in math or physics. "Semantic Field" is originally a term in linguistics, which refers to a lexical set of words grouped semantically, just like the set of embeddings that SAFARI found to be particularly semantically coherent.
* The term "Subspace" here is indeed used in the mathematical sense. However, SAFARI is not limited to discovering sets of semantically coherent embeddings. In Line 9 of the SAFARI Algorithm (Algorithm 1), we perform SVD on the aforementioned set of embeddings to construct a subspace (Algorithm 2) to compute Semantic Shift. The input for computing Semantic Shift is the subspace itself—specifically, the row space and singular values derived from SVD. Therefore, the subspace is a fundamental component of our algorithm.
* The Semantic Field Subspace is not merely a set of vectors but an actual subspace within the embedding vector space. This subspace generalizes the semantic meaning beyond the individual vectors, as evidenced by our experiments, which demonstrate that the subspace itself encapsulates the semantics effectively.

**4. Unclear Experiment Setting**

* The settings shared across all experiments are detailed in Section 5.1, lines 355 to 364. To avoid redundancy and due to page limitations, we did not repeat these settings in each subsection for individual experiments. However, we understand that this way may have made the settings less visible to readers.
* We will refine this section in the next revision to ensure clarity and accessibility. To address the specific comments:
    *  Training: SAFARI does not account for initial embedding training, and the training time in Figure 7 is not for initial embedding training as well.
    *  Embedding: Except the embeddings used by BERT, other embeddings in the experiments come from the Entity Linking model BLINK.

---

### Meta-Review · Area_Chair_tJr1 · 2024-12-20

**Metareview:**

Summary:
The paper introduces the concept of Semantic Field Subspaces and the SAFARI algorithm, which leverages hierarchical clustering and semantic shift computations to uncover hierarchical semantic structures within embedding spaces. The authors propose a method to approximate semantic shift computations and evaluate on five datasets to reveal interpretable and hierarchical semantic structures.

Strengths:
- Important problem of understanding embeddings

Weakness:
- All the reviewers raised major concerns with the paper.
- Experiment Clarity: Several reviewers noted that the experimental settings were unclear, including details about baselines, hyperparameters, and dataset construction.
- Comparison Limitations: The lack of strong baseline comparisons, especially with models like RoBERTa, Electra, and XLNet, weakens the empirical evaluation.
- Theoretical Justifications: Certain mathematical claims, particularly around semantic shift definitions and approximations, need clearer justifications.
- Presentation Issues: Inconsistent notations and unclear definitions were cited as points of confusion.

Decision:
While the work shows promise and addresses an important problem, unfortunately, the paper can't be accepted in its current form and addressing all the concerns would warrant another round of reviewing.

**Additional Comments On Reviewer Discussion:**

We thank the authors and reviewers for engaging during the discussion phase towards improving the paper. Below are some of the highlights:

1. Confusion about the core motivation and relationship to clustering
- Authors clarified SAFARI is not primarily a clustering algorithm but rather uses clustering as a tool for semantic analysis
- Reviewers generally accepted this clarification though some concerns remained

2. Mathematical notation and terminology issues
- Authors acknowledged issues and promised clearer explanations
- Offered detailed explanations of key concepts like Semantic Field Subspace
- Most reviewers found the explanations satisfactory

3. Limited experimental evaluation
- Authors proposed adding more baselines (RoBERTa, Electra, XLNet)
- Suggested additional experiments with SVM integration
- Reviewers remained concerned about anecdotal evidence in some experiments

4. Implementation details and setup
- Authors provided missing details about training procedures and embeddings used
- Clarified SAFARI does not account for initial embedding training
- Better explained experimental configurations

In summary, though the authors provided good clarifications in rebuttal, many suggested improvements would require substantial revision beyond what's possible in the rebuttal period.

---

### Decision · Program_Chairs · 2025-01-22

Reject